# Human-triggered magnification of erosion rates in European Alps since the Bronze Age

William Rapuc [1,2] ✉, Charline Giguet-Covex [1], Julien Bouchez [3], Pierre Sabatier [1], Jérôme Gaillardet [3,4], Kévin Jacq [5], Kim Genuite [6], Jérôme Poulenard [1], Erwan Messager[1] & Fabien Arnaud [1]

A major feature of the Anthropocene is the drastic increase in global soil erosion. Soil erosion is threatening Earth habitability not only as soils are an essential component of the Earth system but also because societies depend on soils. However, proper quantification of the impact of human activities on erosion over thousands of years is still lacking. This is particularly crucial in mountainous areas, where the highest erosion rates are recorded. Here we use the Lake Bourget catchment, one of the largest in the European Alps, to estimate quantitatively the impact of human activities on erosion. Based on a multi-proxy, source-to-sink approach relying on isotopic geochemistry, we discriminate the effects of climate fluctuations from those of human activities on erosion over the last 10,000 years. We demonstrate that until 3800 years ago, climate is the only driver of erosion. From that time on, climate alone cannot explain the measured rates of erosion. Thanks to an unprecedented regional paleoenvironmental reconstruction, we highlight that the development of pastoralism at high altitudes from the Bronze Age onwards and the extension of agriculture starting in the Middle Ages were key factors in the drastic increase in erosion observed in the Alps.

Although one of the major current environmental issues, soils degradation is still too often overlooked[1]. Yet, among other amenities, soils provide the substrate for 95% of human food and constitute the most efficient $CO_2$ sink at Earth's surface[1–3]. Soils are hence a vital resource for human societies, nowadays facing severe degradation due to the increasing extent of cultivated and urbanised lands, which leads to a loss in soil ecosystem functions[4,5] and a dramatic rise in global erosion rates[6–8]. The deleterious potential of soil erosion and the corresponding lack of knowledge are such that Europe and the United Nations have recently called for an improved quantitative evaluation of soil loss over large spatial and long temporal scales[2,9]. Ideally, such assessment should in particular include a quantification of the effects of various factors driving erosion, in order to model their evolution and improve resource management policies[2,6]. Therefore, several

research groups have carried out modelling experiments at European and global scales to draw a picture of the current[9,10] and future[3] states of erosion rates. The prominent role of human activities in soil degradation and global erosion rise was thereby evidenced[6,7,10]. However, to date a critical knowledge gap exists as to when agriculture and grazing became the most significant driver of erosion, due to the compounding effects of climate shifts over the Holocene period; nor has it been possible to quantify accurately the effects of land use on erosion beyond decades or, at most, a few centuries.

Lake sediment sequences host a wealth of information to reconstruct the response of soils to climate and land use over several centuries to millennia[11–13]. In lake sediment studies, sedimentation rates[12] or detrital particle fluxes[14,15] from the corresponding catchments are commonly used as proxies in that aim. Although such data are

[1]EDYTEM, CNRS, Université Savoie Mont Blanc, 73000 Chambéry, France. [2]Department of Earth Sciences, Durham University, Durham DH1 3LE, UK. [3]Université Paris Cité, Institut de Physique Du Globe de Paris, CNRS, 75005 Paris, France. [4]Institut Universitaire de France, Paris, France. [5]Laboratoire Commun SpecSolE, Envisol, CNRS, Université Savoie Mont Blanc, 73000 Chambéry, France. [6]UMR PACEA 5199, CNRS, Université de Bordeaux, 33615 Pessac, France. ✉e-mail: william.l.rapuc@durham.ac.uk

informative about changes in catchment erosion rates through time, they do not allow for assessing the relative roles of different forcing factors, such as climate and human activities[16]. Independent reconstructions of land cover and local climate obtained from models[12] or proxy data[17] can help to identify the potential correlations between erosion rates and these forcing factors but do not allow for a quantification of these effects.

Here we apply an original source-to-sink approach[16] to a 10,000-years-long lake sediment archive (Supplementary Fig. 4) representative of the western European Alps (Supplementary Fig. 1). The Alps presents currently the highest erosion rates over the entire European continent[3,9], making it a hotspot of soil loss[18]. In addition, as one of the best documented places on Earth in terms of climate fluctuations[19–23] during the Holocene and of human activities from the Neolithic period[24–27], this region represents an ideal environment to quantify the impact of both climate and human activities on erosion rates. Lake Bourget drains one of the largest alpine catchments (4976 km$^2$) which is particularly well suited for such a study as it hosts a diversity of landscapes in terms of land use, geomorphology, and lithology[28] (Supplementary Fig. 1). The catchment culminates at 4808 m at the top of the Mont Blanc massif, where the presence of glaciers precludes any land use, such that erosion there is largely controlled by climate[29,30]. Moreover, most of this glacier-covered part of the catchment (158.1 km$^2$ situated between 2300 and 4808 m) is underlain by granites and metamorphic rocks[31] with a geochemical signature distinct from those of the sedimentary formations underlying the lower elevation area[32–34] (3803.9 km$^2$ excluding limestone formations; 232–3057 m), where agro-pastoral activities have developed over the last 7000 years to become a major human activity in the region[24,27,35]. This configuration can be leveraged to quantify the role of various driving factors on erosion in the Lake Bourget catchments, given the following assumptions: (i) erosion in the glacier-covered part of the catchment responds only to climate (temperature and precipitation), whereas (ii) erosion in lowermost, non-glaciated regions depends both on climate and land use; and finally (iii) if climate shifts were the sole factor of changes in erosion, they would affect equally any part of the catchment area. Several studies support the first of these assumptions[36–38] showing that in glaciated regions, climate and glacier thermal regime are the two main factors controlling erosion and the only factors explaining the variations on the time scales of our study[36]. Furthermore, no human activity likely to have enhanced erosion has yet been identified on the Mont Blanc massif[21]. The second assumption is commonly accepted and form the basis of all models based on soil loss equation (e.g., RUSLE models)[3,9,10]. To demonstrate the validity of the third assumption in the context of our study area we have chosen a sedimentary sequence long enough to cover periods when anthropogenic effects on erosion were negligible leaving climate as the only controlling factor for the oldest part of the record. Based on these assumptions, we can quantify the evolution of climate- vs. human-driven erosion through time, such that an anthropogenic excess in erosion can be estimated. To ascertain that this excess erosion is indeed linked to agropastoralism, we employ an approach based on a synthesis of indices of agro-pastoral activities at the scale of the western European Alps. Agro-pastoral activities derived from pollen or environmental DNA data from 8 lakes covering an altitudinal gradient in or near the Lake Bourget catchment were synthesised through the calculation of a site-specific index of "agro-pastoral activity". This index is blind to other human activities such as deforestation and mining. Because each altitudinal range presents its own environmental trajectory over the Holocene[24], these indexes were then binned by altitudinal ranges, that are named foothills (250–850 m), mid-altitude (850–2000 m) and high altitudes (>2000 m). Our source-to-sink approach involves the collection and measurement of 29 samples of fine river sediments from the flood plain of the Rhône and the Arve rivers, the main tributaries of Lake Bourget, and their tributaries (Methods). Eighty-three samples

were also collected in the sediment sequence retrieved in the depo-center of Lake Bourget (Methods). Erosion flux was estimated from the sediment sequence by the computation of the sediment dry bulk density, the sedimentation rate, and the proportion of detrital silicates. Isotopic ratios of neodymium, as well as major and trace element concentration, were measured on the entirety of these samples. These radiogenic isotope systems are commonly used to trace the sources of erosion products[39,40] and due to the different geological nature of glaciated and non-glaciated regions, isotopic ratios of neodymium offer the possibility to track the respective contribution of each region to erosion through time, based on the use of a geochemical mixing model (Fig. 1, Methods).

## Results and discussion

### Recording the erosional effects of climate and land use in the Alps

Here, we obtain two erosion signals (Fig. 2) corresponding to the glaciated region and to the rest of the Lake Bourget catchment, respectively. Erosion signals are expressed as Sediment Yield (SY) value (t.km$^{-2}$.yr$^{-1}$) that yields only relative information on erosion rates in sediment source regions as (i) sediment deposition may occur on floodplains during transport to the lake and (ii) the Rhône River, which supplies sediment to Lake Bourget, only does so while flooding, i.e. during a few days each year[41]. Therefore, no data comparison is possible with other $SY_i$ values such as those obtained from soil erosion models[9,42], such that only variations through time and relationship between erosion rates in the two parts of the catchment is discussed in the present study. Thanks to a precise age-depth model (Supplementary Fig. 4), with a median uncertainty of 60 years and a maximum uncertainty of no more than 100 years over the last 6700 years, it is possible to accurately describe the evolution of these erosion signals over time. Both erosion signals remain low and relatively constant from 9440 to 4880 yr cal BP (Fig. 2a, b). A first increase in erosion rate is observed for both regions from 4,880 to 4,065 yr cal BP. During this period, called the Neoglacial period[43], the Alps experienced a transition towards a wetter and colder period compared to the Early Holocene[44]. This change in climate led to glacier advance at high altitudes[44–46]. The first phase of increase in the erosion signal being similarly observed in both regions is therefore linked to the effect of one of the main climatic transitions of the Holocene in the European Alps. From 3860 yr cal BP onwards, the two regions show different trends, with an increasing erosion rate in the non-glaciated part of the watershed, while the rate remains constant in the glaciated region with the exception of three periods listed hereafter, which corresponds to known phases of glacial advance in the area[21,43]. Indeed, during the cold periods of the Migration period (1700 to 1550 yr cal BP), the Early Middle Ages (950 to 700 yr cal BP), and the end of the Little Ice Age (280–140 yr cal BP), the two regions show a similar increase in erosion or an excess of erosion from the glaciated area (Fig. 2a, b). Over the last thousand years, the intensity and variability of the erosion signal in the glaciated region has far exceeded anything previously recorded (Fig. 2b). This can be explained by the fact that never in recent millennia have the region's glaciers had such a volume and covered such a large area[21,47]. This means that the region's climate is wetter and colder than during the rest of the Holocene[23,48], favouring erosion as well as the massive advance of the glaciers, which will also reinforce the erosion of the area.

### First effects of human activities on erosion 3800 years ago

To quantify the effect of human activities on erosion, the effect of climate fluctuations must be isolated first. The correlation between the erosion signals from the two regions for the Early Holocene (Fig. 2a, b) seems to indicate that before 3860 yr cal BP climate alone was responsible for the erosion of the entire catchment. Indeed, a significant linear relationship (r = 0.86, $p$-value > 10$^{-7}$) can be

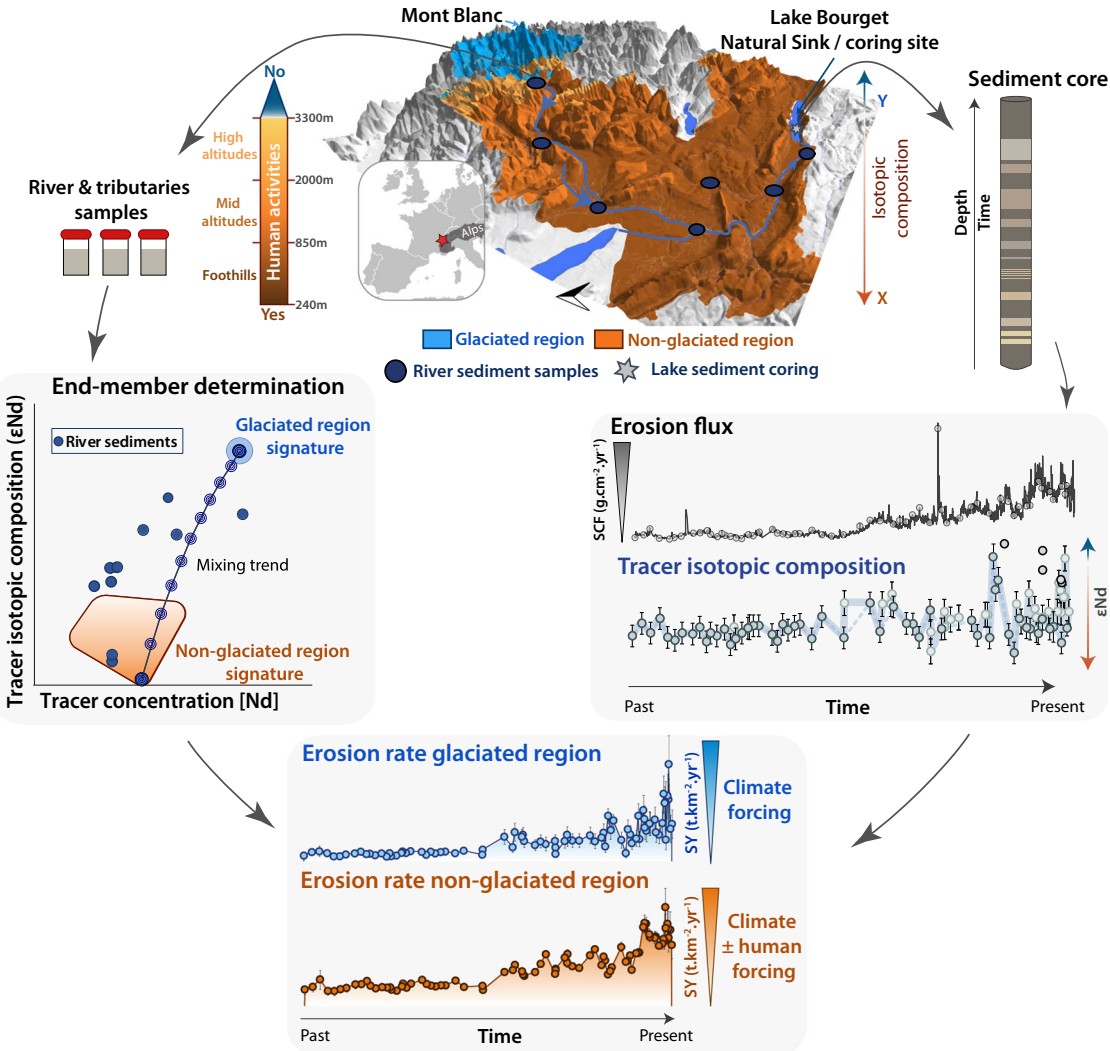

**Fig. 1 | Conceptual model, presenting a 3D-view of a catchment adapted to the approach of the present study.** Example here of Lake Bourget catchment (see Methods), hosting two regions, each of which is characterised by a specific geochemical signature, and in which erosion of one of the regions is influenced mostly by climate fluctuations. To obtain an erosion signal for each of the two regions, isotopic measurements on sediment samples from the source regions, as well as on the lake sedimentary sequence, are needed. These data are used to build a mixing model to obtain the contribution of each region to the erosion signal recorded in the lake. Combined with the knowledge of the total erosion flux in the whole catchment, it is therefore possible to obtain an erosion signal per region over time. The signal from the glaciated region (in blue) corresponds to climate-driven erosion only, while the second one (brown to orange shading) is influenced by both climate and human activities (see Methods for detail).

identified between 9440 and 3860 yr cal BP (Methods). This observation leads us to hypothesize the existence of a linear relationship between the effect of climate on erosion in both glaciated and non-glaciated regions in the European Alps for the Early Holocene. This observation is valid both for a hot, dry period when local glacier extension is very limited (Early Holocene, 9440–4800 yr cal BP) and for a period when the area returns to a wetter, colder climate (Neoglacial, 4800–3800 yr cal BP)[23,28,44–46,48]. This validates the third of our initial assumptions, according to which if climate change were the only factor in the evolution of erosion, it would affect any part of the catchment area in the same way. The differing trends between the two erosion signals from 3860 yr cal BP onwards indicate the appearance of a new driver of erosion in the non-glaciated part of the catchment, superimposed on climate. From that linear relationship an erosion rate can be computed for the non-glaciated region, that should reflect only the effect of climate on the erosion for the rest of the Holocene (Supplementary Fig. 7 and Supplementary Method 6). The ratio between this erosion rate expected from climate effects alone, and the measured erosion rate

corresponds to the excess in erosion in the non-glaciated region, which is not explained by climate fluctuations. Variations in the signal of erosion excess that do not exceed the variability observed over the correlation period (9.440–3.840 yr cal BP) will not be discussed, as such variations may be linked to the noise inherent in the archive used.

From 3800 yr cal BP onwards, climate alone cannot explain the erosion rates recorded in the Lake Bourget catchment. Because it is impossible that large-scale tectonic forcing could have affected only one of the two regions, we attribute the excess in erosion to human activities. The increase in the impact of agro-pastoral activities occurred steadily in the Bronze Age (Fig. 2c) before leading to a first occurrence of a high value of erosion excess at the end of the Bronze Age (3275 and 3025 yr cal BP). The other high excess erosion values are observed during: (i) the transition between Iron Age and Roman Period (2020 yr cal BP); (ii) at the end of the Early Middle Age (1350–1230 yr cal BP), at the end of the High Middle Age (750–710 yr cal BP), and at the end of the Early Modern (215 yr cal BP). A decrease is observed between 615 (1335 CE) and 490 yr cal BP (1460 CE), which, considering the

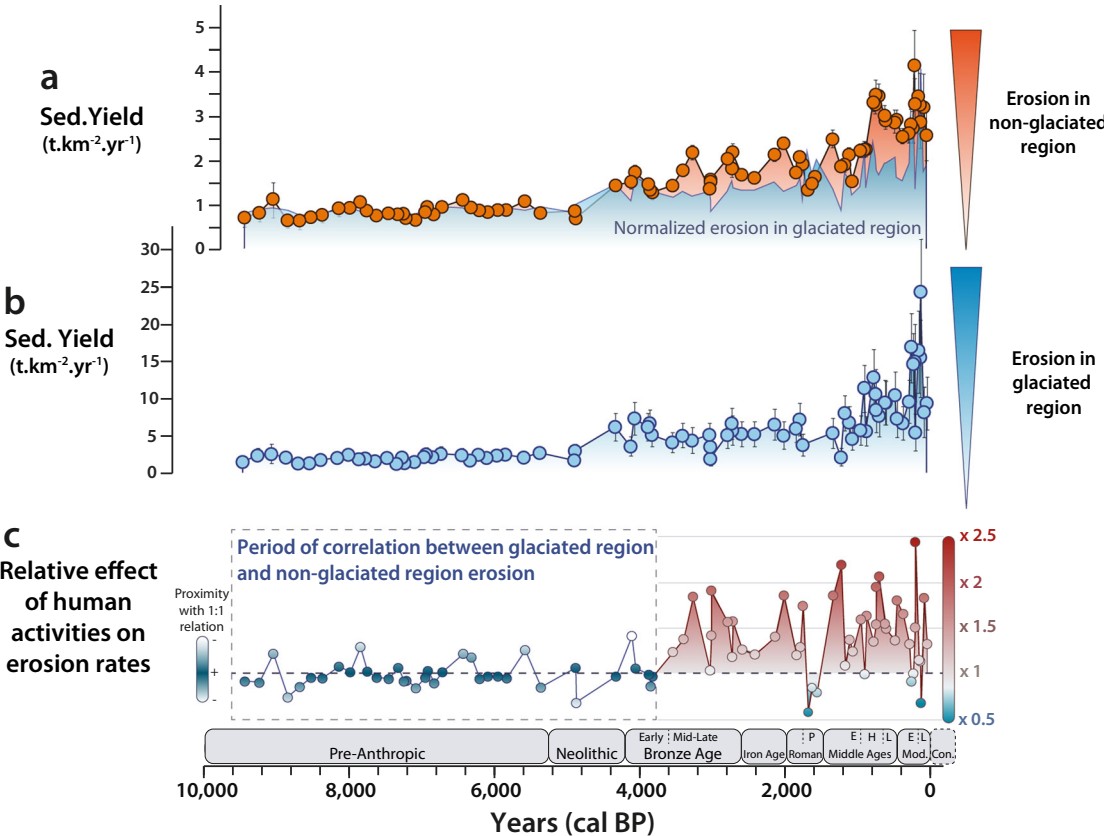

**Fig. 2 | Erosion signals for non-glaciated and glaciated regions and the relative effect of human activities on erosion rates in Lake Bourget catchment.** Sediment yields (expressing erosion rates) for both the (**a**) non-glaciated (in orange) and (**b**) glaciated regions (in blue) of the Lake Bourget catchment, and (**c**) the effect factor of human activities on erosion in the non-glaciated region (blue to red shading), computed as a ratio between the measured erosion rate and the expected erosion rate in the absence of human impact (Supplementary Method 6; Supplementary Fig. 7). These two erosion signals were computed from the detrital silicate flux, the mixing model, the εNd data obtained on both the sedimentary sequence and the river sediments, and by calculating the areas of the glaciated and non-glaciated region in the catchment (Method). The error bars correspond to the standard deviation obtained using a Monte Carlo approach (Method). Colour shading of erosion signals in glaciated and non-glaciated regions highlights increases in intensity compared with early Holocene values. On panel (**a**), the erosion signal for the glaciated region has been normalised to the correlation period values to highlight periods where erosion rates are dissimilar between the two regions. On panel (**c**), the magnification scale represents the effect of human activities on erosion compared with the theoretical value calculated for erosion impacted solely by climatic fluctuations. Periods: Pre-anthropic (until 5250 yr BP), Neolithic (5250–4200 yr BP), Early Bronze-Age (4200–3600 yr BP), Mid to Late Bronze-Age (3600–2600 yr BP), Iron Age (2600–1980 yr BP), Roman Period (1980–1750 yr BP), Post (P) Roman Period (1750–1474 yr BP), Early (E) Middle Age (1474–950 yr BP), High (H) Middle Age (950–650 yr BP), Late (L) Middle Age (650–458 yr BP), Early (E) Modern (458–161 yr BP), Late (L) Modern (161–4 yr BP), Co: Contemporary (4 yr BP – present). Source data are provided as a Source Data file.

uncertainties of the age model, could be explained by the decrease in population after the Black Death[49–51]. Only the Migration Period (1700 to 1550 yr cal BP) and the end of the Little Ice Age (140 yr cal BP; 1810 CE) show excess values below 1. These two periods were characterised by a transition to a colder and wetter climate[19,21] than the average climate of the last two millennia[23]. This had two effects: (i) increasing the surface area covered by glaciers[21,52], (ii) and enhancing erosion in the glacial region through the combined effect of increased precipitation and the advance of the glaciers. These two periods are also known to have seen a marked decline in human activity, linked to invasions during the Migration Period[19,24] and to the reduction in agricultural practices[53,54] probably induced, for both periods by the shortening of summer[24]. These combined effects explain the strong decrease observed in the erosion excess signal for these two periods (Fig. 2c). Regardless of the climatic context, and except for the end of the Little Ice Age (1680 to 1810 CE), agro-pastoral activities have caused a two-to-two-and-a-half-fold increase in erosion rates in the western European Alps (Fig. 2c). Hence, within the climatic context of the western European Alps, agro-pastoral activities have dominated erosion for the last 3800 years, making it necessary to consider human activities as a principal driver of soil erosion over these timescales.

## From indices of agro-pastoral activities to effects on erosion

Any link between human activities and erosion excess remains to be better understood. For this purpose, we have divided the non-glacial region of the Bourget catchment, which is the region affected by agro-pastoral activities, into three parts: (i) the foothills; (ii) the mid-altitude level; and (iii) the high-altitude level (Fig. 1). We hypothesize that the two to three sites selected by altitudinal range illustrate the intensity of pastoral and/or agricultural activities in these areas[24,27,55]. This assumption is considered as robust as the history of activities recorded at the sites in a given altitudinal range are very similar (Supplementary Fig. 8). Because the Lake Bourget integrates the sum of erosion products over all these altitudinal ranges (and is therefore not impacted by local physical and geological characteristics leading to variability in local erosion response), we can assess and quantify the extent of human impact on erosion at such large scale. Using a previously developed methodology[24], we aggregate several tracers of grazing, pastoral, and agricultural activities as recorded by lake sediments into an index of intensity/diversity of agro-pastoral activities. Due to their different natures, each marker of human activity are standardised beforehand. The standardization consists in the normalization to 0–100% of the

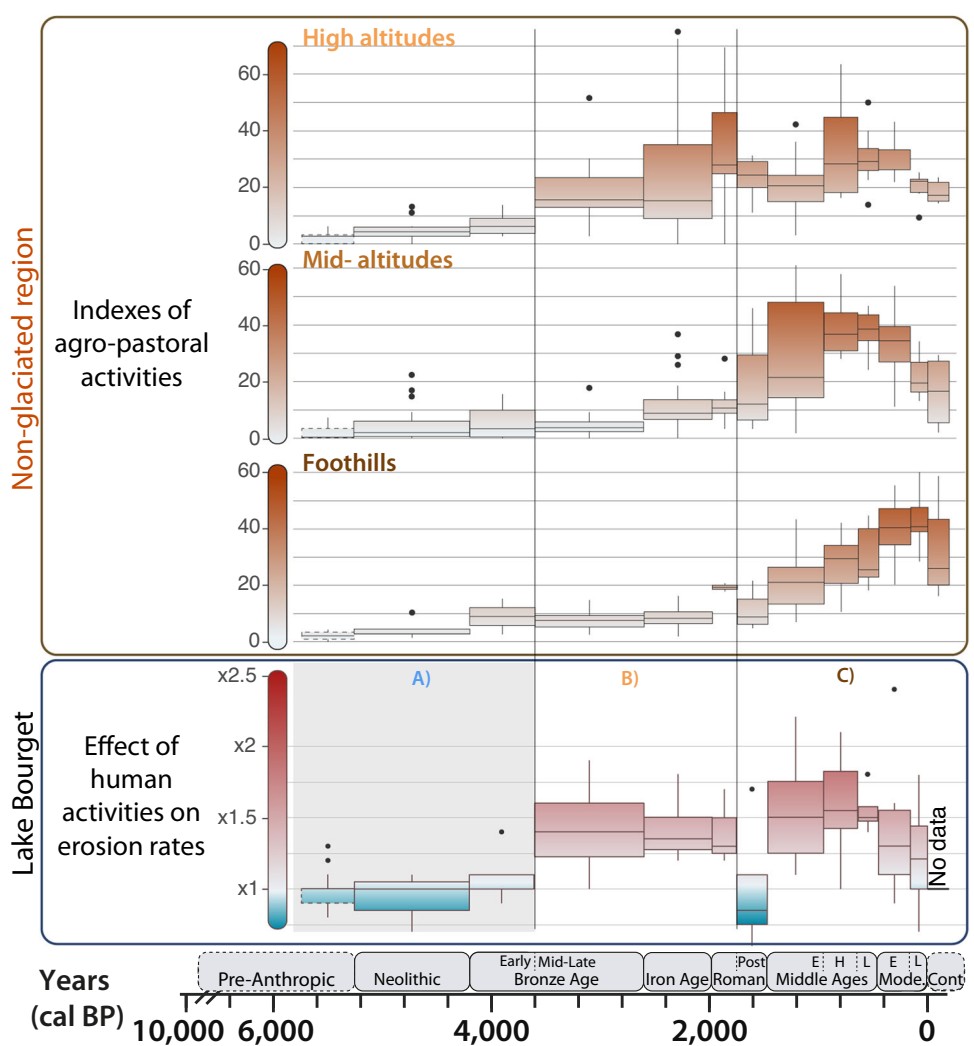

**Fig. 3 | Comparison between index of agro-pastoral activities (brown shading) and the magnification effect of agro-pastoral activities on erosion rates (blue to red shading).** Indexes were computed for three altitudinal ranges of the western European Alps. To facilitate the comparison, the chronicle of the effect of agro-pastoral activities on erosion is presented in the same form as the indexes (Fig. 2). **A** Period used to compute the excess in erosion; (**B**) Period of dominant influence from high-altitude areas; (**C**) Period of dominant influence from foothills/mid-altitudes areas. Source data are provided as a Source Data file.

values of all proxies. Once obtained for each lake, these normalised values are integrated in the calculation of an average index by lake. Finally, these indexes are grouped together by altitudinal range and for each chrono-cultural period (Fig. 3). The integration of data from different lakes and different data-types allows us to offset specific taphonomic issues associated with each proxy and lake. Sedimentary DNA and coprophilous fungi signals are positively affected by climatic or human-triggered erosion, potentially leading to an overestimation of human activities, while pollen counting can be affected by a non-local origin of pollen grains and thus potentially reflect activities in other altitudinal ranges[24]. However, for DNA and coprophilous fungi, because each lake catchment has its own sensitivity to erosion processes, potential biases are not expected to have had an identical pattern from one site to another. The binning of the data by chrono-cultural periods is an additional means of reducing noise in the signals generated by potential taphonomic problems.

The first erosion excess in Lake Bourget is recorded during the Mid-Bronze Age (3600–2600 yr cal BP), a period where agro-pastoral activities increase at high altitudes (Fig. 3). This period corresponds to the arrival of the first flocks of sheep at high altitude[25,56]. The vegetation was sparse, facilitating opening of the vegetation and

establishment of mountain pastures as early as 3800 yr cal BP[51]. Therefore, these data suggest that high-altitude pastoralism has acted as the first factor that has favored soil erosion in the Alps. Then, and until the end of the Roman Empire (1750 yr cal BP), excess erosion remained high but stable, despite an increase of agro-pastoral index activities at high-altitude (Fig. 3). This might be explained by reaching a threshold of soil available to erosion. Until this period, only the high-altitude areas of Lake Bourget catchment seem to host human activities strong enough to impact soil erosion. However, from the Early Middle Ages (1750 yr cal BP) onwards, the excess in erosion increases simultaneously with the evidence of agro-pastoral activities in the mid-altitude and lowland areas. Although high-altitude areas remained characterised by significant pastoralism until the Modern period (460 yr cal BP; 1490 CE), agricultural activities[51], with the advent of the plough from the Early Middle Ages (1750 yr cal BP) onwards[57], seem to have become the main driver of erosion in the western European Alps. It is interesting to notice that even if High (950 to 650 yr cal BP) and Late Middle Ages (650 to 460 yr cal BP) present the highest indices of agro-pastoral activities at mid and high altitude, there is no significant increase in the effect of these activities on erosion. This can correspond to a "system adjustment" phase during which human activities have deliberately (or by chance) evolved to lessen soil erosion, or when

virtually all erodible soil was removed, as previously proposed[24,58]. Except for foothills, the indices of human activity show a decreasing trend from the Early Modern (460 yr cal BP; 1490 CE); probably related to progressive agricultural abandonment of mountain environments[59] leading to a gradual return of forest cover[17,60], which may also have been favored by climate change[61]. This trend is directly reflected in the erosion excess, which decreases until the end of the Late Modern (4 yr cal BP; 1946 CE). Foothills, in contrast, present a differing trend of human activity index towards the Contemporary period (Fig. 3). This suggest that the activities conducted in this area do contribute less in terms of erosion, probably due to more gentle slopes characterizing the foothills reducing both erosion rates and the efficiency of sediment removal.

This study highlights the importance of the retrospective approach for understanding the effect of human activities[11]. Only this type of approach, based on multi-proxy analyses and long timescales, can deliver detailed information on the compounded effects of climate and human activities on soil erosion. Here, we constrain the pace at which agro-pastoral activities led to an amplification of erosion rates in one of the largest Alpine catchment areas over the Holocene. We show that even non-intensive early agro-pastoral activities triggered a direct and major effect on soil erodibility for at least 3800 years. The multiplication of similar approaches could help quantify the impact of human activities on soils at a broader scale, and demonstrate the early impact of human activities on the environment, challenging the hypothesis of an Anthropocene starting with the Industrial revolution[62].

## Methods

### Coring survey, logging, analyses, and dating

In June 2018 and 2019, a total of 56.5 m of sediment was retrieved in the deep basin of Lake Bourget (145 m below lake surface; Supplementary Fig. 2), using 90-mm diameter piston corers (Uwitec, Mondsee, Austria) operated from an UWITEC platform (EDYTEM/LSCE/C2FN). Two-meter-long sections from different holes were taken with a 1-m offset to ensure overlap and provide a continuous record. Sections were split, photographed at high-resolution (20 pixels.mm$^{-1}$), described and logged in detail. Identification of specific layers in the overlapping sections combined with correlations of XRF-core scanner signals allowed the creation of a 13.7-m-long sediment sequence called "LDB18&19" (IGSN n°TOAE0000000005 & n° TOAE0000000006, 45°44.717'N; 5°51.789'E). To characterise the high-resolution variations in major elements throughout the sediment sequence, X-ray fluorescence (XRF) geochemical analyses were performed on the EDYTEM laboratory's Core Scanner (Avaatech XRF Technology). A continuous 5-mm step measurement was applied with a unique run at 10 kV and 0.12 mA for 15 s to detect lightweight elements, such as Al, Si, K, Ca, Ti, Mn, and Fe. The XRF core scanning results are expressed as peak intensity counting (cps). The description of the main sedimentological results is presented in Supplementary Note 1. Ti signal was used to estimate the proportion of detrital silicates in the sediment sequence (Supplementary Method 2). To produce a precise age-depth model for the sediment sequence, we combined identification of historical events such as earthquake and flood deposits with $^{14}$C. Nineteen samples of vegetal macro-organic remains were used to perform $^{14}$C measurements at the LMC14 laboratory (CNRS). Dates were calibrated at 2 sigma using the Intcal20 calibration curve[63], and the age-depth model was performed using the R code package "clam"[64] and "bacon"[65] in R software (Supplementary Method 1). Age-depth model results are shown in Supplementary Fig. 4, Supplementary Table 1.

### Lake and river sediment samples

Fifty-nine 2 cm-thick sediment samples were collected using graduated syringes for geochemical analyses. A constant sampling step of 10 cm was applied where no event deposit interrupted the continuous sedimentation (Source Data). Details of the Sediment dry bulk density (DBD) was measured using sediment samples of an invariant 5 mL volume. To quantify the organic matter and carbonate contents throughout the sediment sequence, loss on ignition (LOI) analyses were conducted following the method developed by Heiri[66] (Supplementary Fig. 3). The age-depth model of the sediment section was performed with "rbacon" and "clam" packages on R software thanks to nineteen samples of terrestrial macro remain (Supplementary Table 1). To characterise the source of sediments accumulated in the Lake Bourget deep basin, 19 river sediment samples were collected between January and February 2018 from flood deposits in all sub-catchments directly contributing to the Arve and Rhône rivers, close to the confluence with the main river (Supplementary Fig. 1; Source Data). Flood deposits were selected as it makes it possible to obtain fine sediment samples covering a long period and representative of each sub-basin. Ten samples were also collected from flood deposits in the main river (Arve & Rhône) at several hundred meters downstream of each confluence to test for the mixing between sub-catchment and main river sediments (Supplementary Fig. 1; Source Data). Sub-catchments covered by this sampling set represent the entire catchment area of the Lake Bourget. Samples were collected manually, and only the fraction finer than 63 μm was used, to ensure that river and lake sediments were comparable.

### Geochemical analyses

Lake and river sediment samples were first dried at 60 °C for 72 h in a laboratory oven and crushed in an agate mortar. Then lake sediment samples were decarbonated by two successive 5 mL HCl 0.5 N leaching steps to remove authigenic and detrital carbonates. All samples then underwent a digestion using concentrated acid mixtures of HF, HNO$_3$, and HCl. Major and trace elements were measured using an Agilent 7900 quadrupole ICP-MS at the PARI analytical platform of IPGP. The reported uncertainties were calculated using the algebraic propagation of blank subtraction and sample count standard deviations ($n = 3$). The NIST®SRM®2709a reference material (San Joaquin soil) was processed as samples and analyzed repeatedly during the sequences to evaluate the accuracy of the measurements. The detection limit was between 0.2 and 0.5 ppt depending on the element, and the internal errors were 5% on average. Nd isotope measurements were all performed at the PARI analytical platform of IPGP on a MC-ICPMS; Neptune (Thermo-Fisher Scientific). After powder digestion, Nd was separated from the sample matrix by extraction chromatography following the same method as in ref. 16. More details on Nd isotope ratio measurements are given in Supplementary Method 3.

### Mixing model

The Nd isotopic composition of the Lake Bourget catchment allows for the identification of two main end members of detrital material (Supplementary Method 4; Supplementary Fig. 6):

(i)  The Mont Blanc Massif (1.A sub-catchment, see Supplementary Fig. 1), where large glacier bodies are concentrated. This area is part of the external massif and presents the highest εNd values (−5.36). We refer to this part of the Lake Bourget catchment as the glaciated region.

(ii)  The rest of the catchment is primarily composed of calcareous sedimentary rocks and Quaternary deposits (Supplementary Fig. 1). In this area εNd values are lower (Source Data). We refer to this part of the Lake Bourget catchment as the non-glaciated region.

We use mean εNd values and [Nd] concentrations of the 1.A and 3.A sub-catchments to track the contribution of the glaciated and non-glaciated region erosion in the total silicate sediment accumulated in

the Lake Bourget, respectively (Source Data). The fractional contribution of each rock source can be estimated using a mixing model:

$$[Nd]^M = [Nd]^A \times f_A + [Nd]^B \times f_B \tag{1}$$

$$f_A + f_B = 1 \tag{2}$$

$$\varepsilon Nd^M \times [Nd]^M = \varepsilon Nd^A \times [Nd]^A \times f_A + \varepsilon Nd^B \times [Nd]^B \times f_B \tag{3}$$

where the superscript M stands for lake sediment ("mixture") values, and the superscripts A and B stand for the source values (glaciated and non-glaciated regions, respectively); $f_x$ represents the fractional contribution of each rock source. Solving this equation system led to estimates of the relative contributions of sources A and B:

$$f_A = \frac{[Nd]^B * \left( \varepsilon Nd^B - \varepsilon Nd^M \right)}{[Nd]^A * \left( \varepsilon Nd^M - \varepsilon Nd^A \right) + [Nd]^B * \left( \varepsilon Nd^B - \varepsilon Nd^M \right)} \tag{4}$$

$$f_B = \frac{[Nd]^A * \left( \varepsilon Nd^A - \varepsilon Nd^M \right)}{[Nd]^B * \left( \varepsilon Nd^M - \varepsilon Nd^B \right) + [Nd]^A * \left( \varepsilon Nd^A - \varepsilon Nd^M \right)} \tag{5}$$

A Monte Carlo method was used to compute uncertainties in the fractional contributions. Random sampling within the analytical uncertainties of each parameter was executed using a Box–Muller transform[67] for each of the 5000 iterations of the Monte Carlo procedure on Excel software. From the simulated parameter distributions, D84, D16, median and standard deviation were obtained for each proportion $f_x$ (Source Data). The same procedure was repeated taking into account uncertainty in the definition of the end member composition. Mixing model results are displayed in Supplementary Fig. 6.

### Computation of the sediment yields
To obtain a quantitative assessment of catchment-scale erosion rates for each region in Lake Bourget catchment, river loads, and corresponding sediment yields were estimated following the methodology of ref. 13. First, the total amount of sediment stored in the Lake Bourget per amount of time ($Stock_{sed}$, in g.yr$^{-1}$) is estimated:

$$Stock_{sed} = SCF \times Lake\ area \tag{6}$$

with $SCF$ corresponding to the siliciclastic flux (g.cm$^{-2}$.yr$^{-1}$), which represents the accumulation of detrital material in the lake Bourget over the Holocene recorded. This flux is obtained from the multiplication of the sedimentation rate ($SR$, cm.yr$^{-1}$), the dry bulk density ($DBD$, g.cm$^{-3}$) and the proportion of detrital silicates ($PDS$, fraction of the total sediment accumulation):

$$SCF = SR \times DBD \times PDS \tag{7}$$

Details about $DBD$, $PDS$ and $SCF$ computation are available in the Supplementary Method 2. $SCF$ results are displayed in Supplementary Fig. 5. As $SR$ varies spatially over the surface of the Lake Bourget[68] we consider only the fraction of the total lake area where $SR$ should be commensurate with that measured at the coring site. This area (*Lake area*, in km²) is estimated at 14.4 km² and corresponds to the area of the lake with depths deeper than 110 m b.l.s[68]. The sediment stock is then multiplied by the contribution of each source ($f_x$, see above) obtained from the mixing model to calculate changes in the source-

specific sediment loads ($SL$, in t.yr$^{-1}$) over time:

$$SL = Stock_{sed} \times f_x \tag{8}$$

To obtain a region-normalised sediment yield value for each source ($SY_i$), SL is normalised to the drainage area of each rock type (*Source area*, in km²):

$$SY_i = \frac{SL}{Source\ area} \tag{9}$$

with $SY_i$ expressed in t.km$^{-2}$.yr$^{-1}$. Details on source area computation are given in Supplementary Method 5.

### Identification of the expected effect of climate on erosion
Identified periods of correlation between erosion signal of both glaciated and non-glaciated regions help to obtain a relationship between both signal for the Early Holocene (Supplementary Fig. 7). To that end, we used the "zoo" package and the "rollapply" function of the R software to examine whether or not the erosion signals from the glaciated and non-glaciated regions are correlated. We produced rolling correlations with sample widths ranging from 3 to 17. The sliding correlation curves between 7*7 and 15*15 roll correlations were then averaged (Cmean) and extracted using Matlab software. Cmean was then derived with a Savitsky Golay polynomial derivative with a sliding window of 5 points and a second order polynomial filter[69]. From the derivative of the signal, peaks are estimated which gives inflection points corresponding to the breakpoints in the correlation relationship of the erosion signals of both regions. Those breakpoints represent the major shifts in the relation of the correlation in erosion between glaciated and non-glaciated regions. More detail is provided about the results of the rolling correlation and the derived linear relationship in the Supplementary Method 6 and in Supplementary Table 2.

### Index of agro-pastoral activities
The proposed index of human activity intensity is the aggregation of all proxies of agricultural and pastoral activities from lake sediment DNA analyses (targeting plants and animals) and/or pollen and coprophilous fungi spore analyses. The list of taxa included for each lake is presented in the Supplementary Table 3. Because these selected proxies are produced by different methods and expressed in different units (i.e., in number of positive replicates for DNA analyses, in percentage for pollen analyses and in number of spores per cm³ for coprophilous fungi), they are normalised to 0–100%. In detail, this standardization procedure consists of changing the highest value of each proxy to 100% and calculating the other values proportionally. Then, the synthetic index is obtained by calculating the mean of the normalised proxies. We integrated the data from several study sites for each altitudinal range (Supplementary Fig. 8). The robustness of this index is underpinned by several hypotheses. First, the integration of proxies from different methods allows for reducing potential biases caused by taphonomic or methodological issues (e.g., *Sporormiella sp.* spores is a more sensitive proxy to the record of low-intensity activities, than animal DNA is[70]). Second, the integration of plant taxa with different ecological preferences related to the intensity of activities allows us to better capture these changes in intensity (e.g. *Plantago sp.* for extensive and intensive activities and *Rumex sp.* for intensive activities and/or the presence of resting places). Third, the integration of several study sites by altitudinal range, which in addition show similar temporal evolution (Supplementary Fig. 8), allows for more reliable reconstructions. In order to synthesize these data and to better highlight the general trends for each altitudinal zone, they are represented via box plots, using "ggplot2" and "grid" packages and "geom_boxplot" function on R software, per cultural period (Source

Data). Finally, this index considers that the more diversified the activities, the greater the intensity of the activities. One must keep in mind that, in regard to assessing human impact on soil stability, this index does not consider deforestation due to difficulties in discriminating human- and climatic-induced forest cover loss on sites around the tree line. Therefore, we contend that the intensity of human activity reflected in our index is underestimated for some periods, especially from the Early/High Middle Ages in foothills and mid-altitude regions. To improve data comparison, the excess erosion signal was also displayed as a box plot for each chrono-cultural period. If we consider that a sample is 2 cm thick, it therefore covers a period that depends on the sedimentation rate. Considering the error in the sedimentation rate, 2 out of the 83 samples span two chrono-cultural periods. The excess erosion values for these two samples have therefore been associated with the box plots for the periods they cover. (Source Data).

## Reporting summary

Further information on research design is available in the Nature Portfolio Reporting Summary linked to this article.

## Data availability

The authors declare that the data supporting the findings of this study are available within the paper and its Supplementary Information files. Source data are provided with this paper.

## Code availability

The authors will make available, on request, any computer code or detailed algorithms used to generate the results presented in the article.

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

## Acknowledgements

This work was financed by CRITLAKE project co-funded by EC2CO and AAP Université Savoie Mont Blanc. Parts of this work were also supported by IPGP multidisciplinary programme PARI and by Paris-IdF region SESAME Grant No. 12015903. We wish to thank the support of the entire staff of the C2FN-DT-INSU associated to the CLIMCORE project for the coring survey. $^{14}$C analyses were acquired thanks to the CNRS-INSU ARTEMIS national radiocarbon AMS measurement programme at Laboratoire de Mesure $^{14}$C (LMC14) in the CEA Institute at Saclay (French

Atomic Energy Commission). We are grateful to Pierre Burckel for help with Q-ICP-MS measurements; to Laëticia Faure for her valuable help and assistance with lab work; to Pascale Louvat, Thibaud Sondag, and Barthélémy Julien for assistance with MC-ICP-MS measurements; and to Raphaël Gallet, Bernard Fanget, Mathilde Banjan, Maude Biguenet, Matthieu Baril, Kim Genuite, Claire Blanchet, Kévin Jacq, Camille Girault, Manon Bajard, Ondřej Bábek, Qi Lin, Xiaqing Wang, Emmanuel Malet for the help during the different coring surveys and Jacques Mourey for help in the field for river sample collection. We wish to thank Damien Guinoiseau for helpful conversations about the interpretation of isotope data. Finally, we would like to thank Carole Bégeot and Elise Doyen for providing us with the pollen data for Lac de Paladru.

## Author contributions

W.R., J.G., P.S. and F.A. designed the study. W.R. wrote the first draft of the paper. W.R. conducted the sedimentological and geochemical analyses of the Lake Bourget sediment sequence. W.R. and J.B. realised the mixing model and conducted statistical analyses on geochemical data. W.R. and C.G.-C. computed the agro-pastoral activities index. W.R. and K.J. developed the method of the human activities effect calculation. W.R. and K.G. produced GIS data on the Lake Bourget catchment. W.R., C.G.-C., J.B., P.S., J.G., K.J., K.G., J.P., E.M. and F.A. contributed to interpreting the results. W.R., C.G.-C., J.B., P.S., J.G., K.J., K.G., J.P., E.M. and F.A. contributed to framing and revising the paper.

## Competing interests

The authors declare no competing interests.
