## [Peer Review File · Nature Communications]

Human-triggered magnification of erosion rates in European Alps since the Bronze AgeREVIEWER COMMENTS

Reviewer #1 (Remarks to the Author):

This manuscript reports a Holocene reconstruction of erosional sediment yield for the European Alps and applies a neat, multi-proxy approach to discriminate between climate and anthropogenic drivers of intensifying soil erosion. I was really pleased to read and review this submission because of its interest and importance. This type of study is welcomed and indeed has been long overdue, in my opinion. So many recent papers exploring Holocene landscape evolution have had to conclude that disentangling the roles of human activity from climate forcings remains a key challenge. Here, the authors have devised an approach that seems to achieve this, which is really commendable. I have limited experience working with neodymium isotopes but, within that caveat, the methods seem appropriate. A number of assumptions are made at each stage of the analysis but, to me, these seem justifiable. There are a handful of moderately substantive areas of clarification that I suggest are necessary prior to publication. I think these will result in a clearer narrative for the broad journal audience and present a more convincing analysis. A few minor comments are also raised.

Main comments:

- 1) Lines 73 – 79: The methodological approach depends on these assumptions holding true. From my modest knowledge of anthropogenic and climate histories from the European Alps over the Holocene, much of which comes from the authors' extensive work in the region, I suspect the assumptions probably are entirely valid. But a reader with less subject knowledge, and given the broad readership of the journal, may be less convinced. I therefore suggest the authors offer some supporting evidence for why these assumptions are justified. This text could, perhaps, be an additional section in the Supplemental Information.
- 2) 'Human activities' are not explicitly defined or described. Some relevant references are provided but, with the broad readership of the journal in mind, I suggest the authors elaborate on ways in which human activities have impacted the land surface over recent millennia that intensify soil erosion and lead to shifts in sedimentological signatures that are detectable in lake sediment cores. It is not until the 'agro-pastoral' section that the reader fully appreciates what types of human activities are being considered. I also note that the authors state on Lines 423-426 that forest cover loss is not being examined. This is a vital statement that ought to be more visible.
- 3) Calculating sediment yields: as mentioned in my introduction, I note a number of assumptions are made at each stage of the analysis. This is inevitable in a study of this nature and they are generally convincing. I do suggest that the authors move the statement on Lines 390 – 395 into the main text. I suspect many readers will overlook this crucial point that calculated sediment yields are relative and therefore should not be compared to catchment sediment yields calculated for different regions or using different methods.
- 4) Explain the full methodology in more detail from the outset. I think the work to infer the impact of agro-pastoral activities from multiple lakes at different elevations is fascinating but it appears rather out of the blue in the manuscript. It is mentioned briefly in the abstract (Lines 21-23) but in quite generic terms. I didn't know what was meant by "an unprecedented regional palaeoenvironmental reconstruction". At the very least, I think some context needs to be provided somewhere between Lines 73 and 89. This also links to Comment #2 on defining human activities. From Line 144, the authors are using agro-pastoral activities as synonymous with 'human activities', which is the term used throughout up to this stage. Some earlier explanation and definition would therefore offer important clarity.
- 5) Naming of periods: Personally, I found it difficult to follow the usage of so many named periods rather than using dates. I appreciate each period is defined in the Figure 2 caption but I needed to continually refer back to this caption when reading the 'Agro-pastoral' section. It was hard work. I suppose I'm unclear what benefit is derived from using such a naming convention as opposed to referring to the same intervals by their years? And using years would likely to be more accessible to readers around the world? I also question whether so many narrow periods are needed? Would the overall interpretation of Figure 2 and Figure 3 change if, say, the boxplots in Figure 3 were wider

because longer time intervals were interpreted? The authors use three primary sub-divisions of time, after all.

Graphs:

Figure 1: An impressive schematic! There is a lot going on here but, in my opinion, it does illustrate the analytical steps. Some minor recommendations: (i) A reader with limited experience of palaeolimnology or source-sink tracing might not follow the use of colour in the river sediment and lake core stratigraphies and how these layers might map (or not) onto the isotopic data; (ii) to be absolutely clear on how climate and human forcings will be discriminated, the authors could consider shading the time series of the "glaciated region" onto the "Erosion rate non-glaciated region" graph. This way the (presumed) human contribution would appear above the background climate contribution; (iii) Would it be possible tilt the DEM so that more of the surface area is visible? (iv) I'm a bit unclear on the orange shading. On the DEM, it looks like one solid shade of orange denoting the 'non-glaciated region'. Is this area meant to be shaded according to the elevation chart on the immediate left, with three shades of brown/orange? This would be very wise – and would provide useful context for the 'agro-pastoral' section – but any variance in orange shading is not visible on the DEM on my screen; (v) Personally, I find it odd that the arrow on the time axes points towards the past rather than present. I suppose this captures the perspective going back through time with increasing depth.

Figure 2: The bottom panel works very well. I do wonder, as with point (ii) in my comment on Figure 1, whether the top two panels could be merged more effectively. It is difficult to judge relative changes in slope and amplitude between the orange and blue time series. I appreciate the bottom panel captures these differences well, but perhaps the top two panels could be normalised and overlain so the divergences really stand out? I also suggest the Figure 2 caption should explain how Nd isotopes have contributed to the data presented in these panels.

Figure 3: as per earlier comments, what is gained by having so many individual periods and thus narrow boxplots?

Minor and typographic comments

Line 11: loss of "land" is an odd choice of term. Perhaps better to remove the words in the middle of the sentence so it reads "A major feature of the Anthropocene is the drastic increase in global soil erosion."

Line 14: better to refer to "thousands of years" given the timescale of your study?

Line 25: "environmental stakes" is slightly odd phrasing

Line 29: "ecosystem" rather than "ecosystemic"

Lines 60-61: Does the region experience the highest rates of erosion in Europe today/recent decades or over centennial/millennial timescales? Worth clarifying.

Line 85: I appreciate the Supplemental Information text contains a lengthy explanation of neodymium isotopes but, given these measurements are pivotal to the study, adding 1-2 more sentences here that explain the suitability of this technique would be valuable.

Lines 96-97: "at the origin" is a rather ambiguous. Do the authors mean the neoglacial triggered the increase in erosion?

Manuscript ID: NCOMMS-23-31162-T

Review conducted by:

Dr Daniel Schillereff

Department of Geography, King's College London

daniel.schillereff@kcl.ac.uk

Reviewer #2 (Remarks to the Author):

Comments

I enjoyed reading this paper; it is elegantly simple in concept and well described and illustrated. Soil erosion is critical and its quantification important. The results of the study reveal a significant effect of humans on erosion. While this link has been known for a long while -- qualitatively -- from many sources of evidence, particularly for mountain areas, this study provides a first pass at quantification and should be of broad interest.

The study strategy does involve some major assumptions about the study system, and I have several concerns about this, developed below. I think the assumptions are justifiable, but as a study such as this may well be used as evidence quite broadly (e.g., to justify further legislation) it is important to demonstrate that it is watertight.

Mary Edwards

77 - there is an assumption here that the climate operates on glaciated and unglaciated regions in a consistent fashion. First, the nature of climate change varies over this space, and second the responses of the systems feature different important components. Changes at high elevation may be less obvious at lower elevations, for example, strong melt during a period of hot, dry summers may shift much glacial sediment but runoff at lower elevations may be reduced. It needs to be justified that this assumption can hold for the observational period.

Fig 2 There is a major increase in average flux and variability in the glacial signal in the past 1000 years. While it is the period of the MO, LIA and post-industrial warming, it does seem to be a large shift (~doubling over rest of Holocene), and it might be useful to comment on this. Is this change mediated by lake-based processes, or changes in rivers, for example?

130 et seq. – Fig 2C – This can be described more precisely. It shows an increase over about 200-400 years away from the 1:1 in the mid-late Bronze Age. “Sudden” thus seems an odd word to use, especially as the new cultural practices would have taken time to spread through the catchment. Looking further at the figure, I see high values during mid and late Bronze Age, one high value at the end of the Iron Age/beginning of Roman period, and an extended high during post-Roman and the first part of the early Middle Age; this is not what is described in the text. Please review the description of the graph. If there is disagreement about what it shows, the data may need to be displayed differently.

This brings up the question of the error in the age model, which should be mentioned in the main text so that these quite fine interpretations of highs and lows in Fig 2C can be judged by the reader.

137-138 This explanation is somewhat weak. There is one rather high point in the glacial curve (but see previous comment), and this causes a temporary negative value. What would be suddenly different about vegetation at this point; major vegetation changes over the catchment would take 10s to 100s of years to develop? If anything, naturally reduced vegetation high elevations and extended glacial extents should have worked in the same direction. More likely there was a contemporaneous reduction in agricultural practice (not “in any case”, but one half of the explanation of the odd value). This brings one to thinking about the noise inherent in any proxy data set from lake sediments. A brief discussion of potential sources of noise (e.g., a decadal variation in river floods affecting a given sampling point) would be helpful. Seeing references to flood deposits in the methods makes this even more important to be clear about.

151 et seq. When there is more erosion, more material, including various disturbance proxies, enter the lake (flux rate goes up). If the erosion were caused by deteriorating climate conditions, not increased human activity, it could appear that there had been more human activity when in fact there had not—if erosion caused a greater proportion of available marker proxies to be transported, thus maintaining or increasing concentrations, it will appear that flux goes up. This circularity is not really addressed in ref 24 either. One needs a fairly robust independent climate-change record to circumvent this possible circularity in the argument. The narrative climate-change background used here is, I suspect, the best there is. The LIA peak event may be the best illustration that climate deterioration and massive erosion in the mountains plus climate deterioration in the lowlands is not linked to enhanced lowland fluxes, though the human-disturbance figure shows this less clearly as the time-slices are variable. At any rate, as with other rather large assumptions, this should be addressed in

the main text somewhere.

Minor comments

11 - Better to say ...“loss of soil caused by a drastic increase in erosion”....otherwise one wonders exactly what is meant by “land”, a word that could be interpreted in many ways

13 - use of “the Earth system” is probably more common and clear

16 - to estimate quantitatively

22 - Bronze Age onward and ... (no comma)

25 - unclear what “stakes” means; “current major environmental issues”...?

27 - consistency: Human or human (latter preferable)

28 -- increasing extent?

29 - not sure what “artificialized” means, please clarify

35 -- experiments?

37 -- thereby evidenced

39 -- quantify accurately

40 -- ? beyond decades or, at most, a few centuries?

48 - omit comma

56 - great figure, but in caption, it would be helpful to say which is which - “The signal from the glaciated region corresponds....while that from the unglaciated....”

59 -- 10,000-year

65 -- well suited (no hyphen, WORD is wrong!)

69 -- situated between?

76 -- omit comma

77 - they would affect

81 - fix “sediment sediments”

91 - Phrase “Thanks to our approach” can be omitted. Ok to say “we obtained”..

95 - associated with

103 Figure 2. Colour shading on left doesn't have a scale and just shows in a quantitative way how far a point is from 1:1 ; it might be clearer with just joined dots and no shading, as the difference between the blue zone and the orange zone is obvious. Graphs should be labelled ABC here, as it is important to direct reader to the correct graph in the related discussion.

106 Colour shading on right needs clarification in caption....and the effect of human activities on erosion in the non-glaciated region, computed as a ratio of the measured erosion rate and that expected in the absence of human impact (right-hand scale, bottom graph/Fig 2C). Also “ER” for end of Roman period is confusing, as “E” is elsewhere used to signify “early”; use Post or After?

111 - late Middle age (not latte)

136 - advance of glaciers led to.....

171 - this statement is rather unclear.

Either human activities were adapted to lessen soil erosion, or the limits of erodible soil were reached, as previously proposed?

182 - finesse? Might need to be explained more

Perhaps...“ This study highlights the importance of the retrospective approach for understanding the effect of human activities. Only this approach, based on multi-proxy analyses and long timescales, can deliver detailed information on the compounded effects of climate and human activities on soil erosion.

304 - 1-m offset

306 - in the overlapping

314 - flood deposits

317 - see main text, reader should know about the age-model uncertainty

325 - I think I can see why flood deposits are used, but it would be good to explain it here.

391 - replace 1/ and 2/ with 1) and 2)

398 - “to that sim” ??makes no sense. “To that end”?

404 - shifts in the relation of the correlation in erosion between.....

Reviewer #3 (Remarks to the Author):

This paper constitutes an important source of insights into the evolving landscape of soil erosion within the Alps, which is a prominent hotspot for soil loss in Europe. The authors' extensive work on an approximately 11-meter composite record to estimate erosion rates spanning several millennia is commendable. The meticulous detailing of their methods, including a new interesting source-to-sink approach, adds credibility to their findings. Moreover, the robustness of the age model, spanning roughly 9.4 thousand years before present, is particularly impressive.

I believe that the results presented in this manuscript hold significant value for publication in Nature Communications, particularly due to the previously unrecorded surge in soil erosion that commenced approximately 3800 years ago.

My main concern is the following: The observed increase in erosion rates in both glaciated and non-glaciated regions around 3800 years ago (or 4.2k; see last sentence). In the glaciated region, sedimentation yield clearly demonstrates a step change during this period, as also indicated by ϵNd values. On the other hand, the non-glaciated region exhibits more of a gradual upward trend, but both regions unmistakably indicate an increase around 3.8kBP. Given that human activity did not significantly impact the glaciated region, it seems that more discussions for this step change in sediment yield at 3.8kBP in the glaciated area be warranted. It would be interesting to explore into how the neoglacial cooling, a period marked by declining temperatures, could have affected sedimentation rates in both glaciated and non-glaciated regions. Perhaps a simple insolation curve in Fig.2 would give a better insight on this. Also, looking at Fig. 2's chronology, it makes me wonder if the step change actually occurred around 4.2k, aka the so-called "4.2k event"? As the age model is quite solid at this depth. Please provide more discussions.

Minor comments:

- Is any population size estimates for the different cultural epochs? Please add reference for each of these periods.

- No grain-size analysis is shown in the paper. Has it been done?

Line 81 : sediment sediments

Line 149 : please provide the number of sites

Line 154 : provide more info about the standardization procedure

Line 398 : aim, remove "s"

Line 412 : please provide more info (0-100%?)

Nature Communications manuscript NCOMMS-23-31162-T

REVIEWER COMMENTS

Reviewer #1 (Remarks to the Author):

This manuscript reports a Holocene reconstruction of erosional sediment yield for the European Alps and applies a neat, multi-proxy approach to discriminate between climate and anthropogenic drivers of intensifying soil erosion. I was really pleased to read and review this submission because of its interest and importance. This type of study is welcomed and indeed has been long overdue, in my opinion. So many recent papers exploring Holocene landscape evolution have had to conclude that disentangling the roles of human activity from climate forcings remains a key challenge. Here, the authors have devised an approach that seems to achieve this, which is really commendable. I have limited experience working with neodymium isotopes but, within that caveat, the methods seem appropriate. A number of assumptions are made at each stage of the analysis but, to me, these seem justifiable. There are a handful of moderately substantive areas of clarification that I suggest are necessary prior to publication. I think these will result in a clearer narrative for the broad journal audience and present a more convincing analysis. A few minor comments are also raised.

Main comments:

1) Lines 73 – 79: The methodological approach depends on these assumptions holding true. From my modest knowledge of anthropogenic and climate histories from the European Alps over the Holocene, much of which comes from the authors' extensive work in the region, I suspect the assumptions probably are entirely valid. But a reader with less subject knowledge, and given the broad readership of the journal, may be less convinced. I therefore suggest the authors offer some supporting evidence for why these assumptions are justified. This text could, perhaps, be an additional section in the Supplemental Information.

We thank the reviewer for this comment. Indeed, in the previous version we did not discuss these assumptions. We now provide supporting evidence for the two first assumptions, the last one being tested and confirmed thanks to the results of the present study. These sentences were added directly after the description of the assumptions (lines 82 – 90 of the revised manuscript): *“Several studies support the first of these assumptions (Koppes et al., 2015; Lehmann et al., 2019; Seguinot and Delaney, 2021) showing that in glaciated regions, climate and glacier thermal regime are the two main factors controlling erosion and the only factors explaining the variations on the time scales of our study. Furthermore, no human activity likely to have enhanced erosion has yet been identified on the Mont Blanc massif (Le Roy et al., 2015). The second assumption is commonly accepted and forms the basis of all models based on soil loss equation (e.g., RUSLE models) (Borrelli et al., 2020, 2017; Panagos et al., 2015). To demonstrate the validity of the third assumption in the context of our study area we have chosen a sedimentary sequence long enough to cover periods when anthropogenic effects on erosion were negligible leaving climate as the only controlling factor for the oldest part of the record.”*

We have also added a sentence in the “First effects of human activities on erosion 3,800 years ago” section to explain that the observation of a linear relationship between the effect of climate on erosion in both glaciated and non-glaciated regions in the European Alps for the Early Holocene validates the third of our initial assumptions (lines 166–168): *“This validates the third of our initial assumptions, according to which if climate change were the only factor in the evolution of erosion, it would affect any part of the catchment area in the same way.”*

2) ‘Human activities’ are not explicitly defined or described. Some relevant references are provided but, with the broad readership of the journal in mind, I suggest the authors elaborate on ways in which human activities have impacted the land surface over recent millennia that intensify soil erosion and lead to shifts in sedimentological signatures that are detectable in lake sediment cores. It is not until the ‘agro-pastoral’ section that the reader fully appreciates what types of human activities are being considered. I also note that the authors state on Lines 423–426 that forest cover loss is not being examined. This is a vital statement that ought to be more visible.

We recognize that the term “human activities” was not clearly defined in the previous version of the manuscript. Thanks to this comment and to the comment 4) we now describe in more details the type of human activities that are considered in this study: i.e., agriculture and grazing. As soon as line 40 we replaced “human activities” by “agriculture and grazing”. We now also explain that ore extraction and deforestation associated to any other activities cannot be directly assessed by the selected approach. However, this does not represent a major bias as agro-pastoral activities are the main activities reported in the western European Alps from the Bronze Age. Please, see our answer to comment 4) for more details.

3) Calculating sediment yields: as mentioned in my introduction, I note a number of assumptions are made at each stage of the analysis. This is inevitable in a study of this nature and they are generally convincing. I do suggest that the authors move the statement on Lines 390 – 395 into the main text. I suspect many readers will overlook this crucial point that calculated sediment yields are relative and therefore should not be compared to catchment sediment yields calculated for different regions or using different methods.

Thank you for this helpful comment. The statement on Lines 390–395 was transferred to the main text at the beginning of the section: “Results and Discussion

Recording the erosional effects of climate and land use in the Alps"; after line 112. We also added some information about the uncertainties of the age-depth model according to the comment of Reviewer #2: *"Erosion signals are expressed as Sediment Yield (SY) value ($t.km^{-2}.yr^{-1}$) that yields only relative information on erosion rates in sediment source regions as i) sediment deposition may occur on floodplains during transport to the lake and ii) the Rhône River, which supplies sediment to Lake Bourget, only does so while flooding, i.e. during a few days each year. Therefore, no data comparison is possible with other SY_i values such as those obtained from soil erosion models, such that only variations through time and relationship between erosion rates in the two parts of the catchment is discussed in the present study. Thanks to a precise age-depth model (Supplementary Fig. 4), with a median uncertainty of 60 years and a maximum uncertainty of no more than 100 years over the last 6,700 years, it is possible to accurately describe the evolution of these erosion signals over time."*

4) Explain the full methodology in more detail from the outset. I think the work to infer the impact of agro-pastoral activities from multiple lakes at different elevations is fascinating but it appears rather out of the blue in the manuscript. It is mentioned briefly in the abstract (Lines 21-23) but in quite generic terms. I didn't know what was meant by "an unprecedented regional palaeoenvironmental reconstruction". At the very least, I think some context needs to be provided somewhere between Lines 73 and 89. This also links to Comment #2 on defining human activities. From Line 144, the authors are using agro-pastoral activities as synonymous with 'human activities', which is the term used throughout up to this stage. Some earlier explanation and definition would therefore offer important clarity.

We agree with the reviewer's comment. In order to better explain how we linked the excess in erosion computed to some specific activities we added text at the end of the description of our approach (lines 91 – 99 of the revised manuscript): *"To ascertain that this excess erosion is indeed linked to agropastoralism, we employ an approach based on a synthesis of indices of agro-pastoral activities at the scale of the western European Alps. Agro-pastoral activities derived from pollen or environmental DNA data from 8 lakes covering an altitudinal gradient in or near the Lake Bourget catchment were synthesized through the calculation of a site-specific index of "agro-pastoral activity". This index is blind to other human activities such as deforestation and mining. Because each altitudinal range presents its own environmental trajectory over the Holocene, these indexes were then binned by altitudinal ranges, that are named foothills (250 – 850 m), mid-altitude (850 – 2000 m) and high altitudes (> 2000m)".* Thanks to this comment we have also answered the reviewer's comment 2) by adding direct information of the type of human activities that are considered in this study: agriculture and grazing. Most of the mentions of "human activities" were replaced by the words "agro-pastoral activities" within the main text.

5) Naming of periods: Personally, I found it difficult to follow the usage of so many named periods rather than using dates. I appreciate each period is defined in the Figure 2 caption but I needed to continually refer back to this caption when reading the 'Agro-pastoral' section. It was hard work. I suppose I'm unclear what benefit is derived from using such a naming convention as opposed to referring to the same intervals by their years? And using years would likely to be more accessible to readers around the world? I also question whether so many narrow periods are needed? Would the overall interpretation of Figure 2 and Figure 3 change if, say, the boxplots in Figure 3 were wider because longer time intervals were interpreted? The authors use three primary sub-divisions of time, after all.

We freely admit that reading the last section of the discussion was made complex by the lack of precise dates associated with each period cited, which is why we decided to add the dates or periods covered each time a reference was made to a period directly in the main text. Please see the section "From indices of agro-pastoral activities to effects on erosion" where changes have been applied directly. We think that adding dates and periods directly to the main text will make it easier to read.

We would, however, like to point out that in the context of a publication in a generalist journal such as Nature Communications, and to reach the widest possible audience, it is probably preferable to leave references to chrono-cultural periods in order to also speak to an audience more closely related to archaeology and history. The dynamics of human activities are consistent with the chrono-cultural period breakdown and therefore with the dynamics of societies in the Alps. We believe that it is therefore important, particularly so that this study can be used by historians or archaeologists, to retain at least the division by period as done in the previous version. It is true that using fewer divisions would not affect the results, but it would be more difficult to highlight the two main periods of land use, with pastoralism affecting the higher altitudes before agriculture took over from the Post-Roman Empire period. This would also have the effect of significantly reducing the resolution and level of detail presented in this study. Although the current level of detail is not essential for drawing conclusions from this study, it is highly likely that these data will be used and that the level of detail they afford may present a particular interest.

Graphs:

Figure 1: An impressive schematic! There is a lot going on here but, in my opinion, it does illustrate the analytical steps. Some minor recommendations: (i) A reader with limited experience of palaeolimnology or source-sink tracing might not follow the use of colour in the river sediment and lake core stratigraphies and how these layers might map (or not) onto the isotopic data; (ii) to be absolutely clear on how climate and human forcings will be

discriminated, the authors could consider shading the time series of the “glaciated region” onto the “Erosion rate non-glaciated region” graph. This way the (presumed) human contribution would appear above the background climate contribution; (iii) Would it be possible tilt the DEM so that more of the surface area is visible? (iv) I’m a bit unclear on the orange shading. On the DEM, it looks like one solid shade of orange denoting the ‘non-glaciated region’. Is this area meant to be shaded according to the elevation chart on the immediate left, with three shades of brown/orange? This would be very wise – and would provide useful context for the ‘agro-pastoral’ section – but any variance in orange shading is not visible on the DEM on my screen; (v) Personally, I find it odd that the arrow on the time axes points towards the past rather than present. I suppose this captures the perspective going back through time with increasing depth.

These comments have led to a significant improvement of the readability of this figure. We have therefore (i) simplified the colours of the river sediment samples to avoid any misunderstanding. The point here is not that the reader thinks that colours might play any role in tracing the source of the sediment. (ii) We took the decision not to superimpose the two erosion signals in this figure because the idea of this figure is to show how we manage to isolate the erosion signals from the glaciated and non-glaciated zones. However, we have used this idea in the revised version of Figure 2. (iii) The DEM was tilted according to the reviewer comment such that more of the catchment surface area is now visible. (iv) Thanks to this comment, we realized that it was probably preferable to present the different altitudinal zones of the catchment directly here. We have therefore applied a colour gradient according to altitude and modified the colour of the vertical scale. (v) We have reversed the direction of the arrows.

Figure 2: The bottom panel works very well. I do wonder, as with point (ii) in my comment on Figure 1, whether the top two panels could be merged more effectively. It is difficult to judge relative changes in slope and amplitude between the orange and blue time series. I appreciate the bottom panel captures these differences well, but perhaps the top two panels could be normalised and overlain so the divergences really stand out? I also suggest the Figure 2 caption should explain how Nd isotopes have contributed to the data presented in these panels.

We completely agree with this comment. However, bringing the two erosion signals together on the same axis would mean no longer showing their fluctuations on different scales and potentially hide the uncertainties. We thus decided to add the normalised erosion signal of the glaciated region directly onto the erosion signal of the non-glaciated region to highlight the differences between both signals. Thanks to this modification the observation of trends of both signals is now made easier. Information about this modification is given in the figure caption (lines 149-150): “*On panel a, the erosion signal for the glaciated region has been normalised to the correlation period values to highlight periods where erosion rates are dissimilar between the two regions.*”.

We have also added some more information about the way the Nd isotopic composition helped to obtain those two signals in the Figure 2 caption (lines 145-147): “*These two erosion signals were computed from the detrital silicate flux, the mixing model, the ϵ Nd data obtained on both the sedimentary sequence and the river sediments, and by calculating the areas of the glaciated and non-glaciated region in the catchment (Methods).*”.

Figure 3: as per earlier comments, what is gained by having so many individual periods and thus narrow boxplots?

As we already stated previously in the answer of the comment 5) we think that the dynamics of human activities are consistent with the chrono-cultural period breakdown and therefore with the dynamics of societies in the Alps. Besides being important for a potential use of the data by historians or archaeologists, reducing this figure by presenting only 3 box plots for the 3 main periods (i.e., the pre-anthropoc period, the period from 3800 cal BP to the end of the Roman period, and the post-Roman Empire period to the present day) would not make it possible to see the trends clearly. Indeed, as the variability of the erosion signal in erosion and human activity are significant within the last two periods, these periods would not be statistically different, and it would be difficult to draw conclusions as clearly.

Minor and typographic comments

Line 11: loss of “land” is an odd choice of term. Perhaps better to remove the words in the middle of the sentence so it reads “A major feature of the Anthropocene is the drastic increase in global soil erosion.”

This has been changed accordingly.

Line 14: better to refer to “thousands of years” given the timescale of your study?

We agree and the text was modified according to the reviewer comment.

Line 25: “environmental stakes” is slightly odd phrasing

Thank you for this remark, we modified this into “environmental issues”.

Line 29: “ecosystem” rather than “ecosystemic”

This change has been applied.

Lines 60-61: Does the region experience the highest rates of erosion in Europe today/recent decades or over centennial/millennial timescales? Worth clarifying.

This has been clarified. Indeed, here we meant the current erosion rates, and we modified the text according to this remark. “The Alps presents currently the highest erosion rates over the entire European continent...”.

Line 85: I appreciate the Supplemental Information text contains a lengthy explanation of neodymium isotopes but, given these measurements are pivotal to the study, adding 1-2 more sentences here that explain the suitability of this technique would be valuable.

Thank you for this remark. We recognize that, considering the broad readership of the journal, some detail about the use of neodymium isotopes associated to the source-to-sink approach is needed here. We thus decided to add: *“These radiogenic isotope systems are commonly used to trace the sources of erosion products and due to the different geological nature of glaciated and non-glaciated regions, isotopic ratios of neodymium offer the possibility to track the respective contribution of each region to erosion through time, based on the use of a geochemical mixing model (Fig. 1, Methods).”*. We also added two new references to support our statement.

Lines 96-97: “at the origin” is a rather ambiguous. Do the authors mean the neoglacial triggered the increase in erosion?

Thank you for this remark, we modified “at the origin” into “the main driver” to remove any ambiguity.

Manuscript ID: NCOMMS-23-31162-T

Review conducted by:

Dr Daniel Schillereff

Department of Geography, King's College London

daniel.schillereff@kcl.ac.uk

Reviewer #2 (Remarks to the Author):

Comments

I enjoyed reading this paper; it is elegantly simple in concept and well described and illustrated.

Soil erosion is critical and its quantification important. The results of the study reveal a significant effect of humans on erosion. While this link has been known for a long while -- qualitatively -- from many sources of evidence, particularly for mountain areas, this study provides a first pass at quantification and should be of broad interest. The study strategy does involve some major assumptions about the study system, and I have several concerns about this, developed below. I think the assumptions are justifiable, but as a study such as this may well be used as evidence quite broadly (e.g., to justify further legislation) it is important to demonstrate that it is watertight.

Mary Edwards

77 - there is an assumption here that the climate operates on glaciated and unglaciated regions in a consistent fashion. First, the nature of climate change varies over this space, and second the responses of the systems feature different important components. Changes at high elevation may be less obvious at lower elevations, for example, strong melt during a period of hot, dry summers may shift much glacial sediment but runoff at lower elevations may be reduced. It needs to be justified that this assumption can hold for the observational period.

We completely agree with this comment. Here we were simply making the assumption according to which there should be a relationship between erosion in high-altitude glaciated areas and in the rest of the catchment area. This relationship has not yet been demonstrated in the catchment studied. However, by examining the erosion signals obtained for the glaciated region and the non-glaciated region we were able to demonstrate a linear relationship between the two signals over periods when only climatic fluctuations could have had an impact on erosion. Moreover, this relationship does not only exist at the beginning of the Holocene, a period during which the climate is rather stable, dry, and warm. This relationship can also be seen during the Neoglacial period (4800 - 3800 yr cal BP), when the climate of the European Alps became colder and wetter, and glaciers re-developed. The existence of a relationship between the two erosion signals covering two contrasted climatic periods allows us to validate this hypothesis.

Then, according to this comment and the comment 1) of Reviewer #1 we now better discuss the different assumptions and explain that the assumption of a linear relationship between the glaciated and the non-glaciated region has not been demonstrated yet in the literature but constitutes one of the main results of this study (lines 87-90 of the revised manuscript): *“To demonstrate the validity of the third assumption in the context of our study area we have chosen a sedimentary sequence long enough to cover periods when anthropogenic effects on erosion were negligible leaving climate as the only controlling factor for the oldest part of the record.”*. We have also added a sentence in the section “First effects of human activities on erosion 3,800 years ago” allowing to describe that the observation of a linear relationship between the erosion signal of the two region is true for two kind of climate periods : the dry and warm Early Holocene and the Neoglacial period (4800 – 3800 yr cal BP): *“This observation is valid both for a hot, dry period when local glacier extension is very limited (Early Holocene, 9,440 – 4800 yr cal BP) and for a period when the area returns to a wetter, colder climate (Neoglacial, 4800 – 3800 yr cal BP). This validates the third of our initial assumptions, according to which*

if climate change were the only factor in the evolution of erosion, it would affect any part of the catchment area in the same way.”. We have also added a few references to clarify the climatic fluctuations of the periods under consideration:

- 44. Ivy-Ochs, S. et al. Latest Pleistocene and Holocene glacier variations in the European Alps. *Quat. Sci. Rev.* 28, 2137–2149 (2009).
- 45. Mauri, A., Davis, B. A. S., Collins, P. M. & Kaplan, J. O. The climate of Europe during the Holocene: a gridded pollen-based reconstruction and its multi-proxy evaluation. *Quat. Sci. Rev.* 112, 109–127 (2015).
- 46. Luetscher, M., Hoffmann, D. L., Frisia, S. & Spötl, C. Holocene glacier history from alpine speleothems, Milchbach cave, Switzerland. *Earth Planet. Sci. Lett.* 302, 95–106 (2011).
- 47. Simonneau, A. et al. Tracking Holocene glacial and high-altitude alpine environments fluctuations from minerogenic and organic markers in proglacial lake sediments (Lake Blanc Huez, Western French Alps). *Quat. Sci. Rev.* 89, 27–43 (2014).

Thanks to this comment and the comments of Reviewer #1 we believe that all of our assumptions are now sufficiently well-founded to be used.

Fig 2 There is a major increase in average flux and variability in the glacial signal in the past 1000 years. While it is the period of the MO, LIA and post-industrial warming, it does seem to be a large shift (~doubling over rest of Holocene), and it might be useful to comment on this. Is this change mediated by lake-based processes, or changes in rivers, for example?

It is true that the average increase in the flux and variability of the glacial erosion signal has been very significant over the last 1,000 years. Over this period, no significant changes were observed in the sedimentary processes at work in Lac du Bourget or in the functioning of the rivers in the catchment area. The first changes in sedimentation type, influenced by the eutrophication of the lake, did not occur until the end of the modern period (see Jenny et al., 2013), and we decided not to study this period precisely to avoid any influence of sedimentation changes on our signal. This increase in average flux is consistent with what has been observed in terms of glacier volumes: over the last 4 millennia the Mont Blanc glaciers (as well as other Alpine glaciers) have never been as large and voluminous as over the last millennium. This is explained in the following references:

- Le Roy, M. et al. Calendar-dated glacier variations in the western European Alps during the Neoglacial: the Mer de Glace record, Mont Blanc massif. *Quat. Sci. Rev.* 108, 1–22 (2015).
- Holzhauser, H., Magny, M. & Zumbühl, H. J. Glacier and lake-level variations in west-central Europe over the last 3500 years. *The Holocene* 15, 789–801 (2005).

The increase in variability can be linked to both larger swings in glacier volume, and to the improved resolution of our record for the last millennium.

Following this comment, we now describe more in detail the erosion signal of the glaciated region (lines 134 – 139): *“Over the last thousand years, the intensity and variability of the erosion signal in the glaciated region has far exceeded anything previously recorded (Fig. 2b). This can be explained by the fact that never in recent millennia have the region’s glaciers had such a volume and covered such a large area. This means that the region’s climate is wetter and colder than during the rest of the Holocene, favouring erosion as well as the massive advance of the glaciers, which will also reinforce the erosion of the area”.*

Reference cited in this comment:

Jenny, J-P. et al. A spatiotemporal investigation of varved sediments highlights the dynamics of hypolimnetic hypoxia in a large hard-water lake over the last 150 years, *Limnology and Oceanography*, 58, (2013) doi: 10.4319/lo.2013.58.4.1395.

130 et seq. – Fig 2C – This can be described more precisely. It shows an increase over about 200-400 years away from the 1:1 in the mid-late Bronze Age. “Sudden” thus seems an odd word to use, especially as the new cultural practices would have taken time to spread through the catchment.

Looking further at the figure, I see high values during mid and late Bronze Age, one high value at the end of the Iron Age/beginning of Roman period, and an extended high during post-Roman and the first part of the early Middle Age; this is not what is described in the text. Please review the description of the graph. If there is disagreement about what it shows, the data may need to be displayed differently.

This brings up the question of the error in the age model, which should be mentioned in the main text so that these quite fine interpretations of highs and lows in Fig 2C can be judged by the reader.

Thank you very much for that comment. It made us realise that the description of the excess erosion signal attributed to human activities was not complete. We have therefore modified the paragraph concerned, in line with this comment, to include the periods of the signal that were not previously described. We believe that, as it stands, the description of the signal is complete and accurately reflects what is shown in Fig 2c. According to this comment and one of your minor comments, we also added in the main text information about the age-model uncertainty: *“Thanks to a precise age-depth model (Supplementary Fig. 4), with a median uncertainty of 60 years and a maximum uncertainty of no more than 100 years over the last 6,700 years, it is possible to accurately describe the evolution of these erosion signals over time.”.*

137-138 This explanation is somewhat weak. There is one rather high point in the glacial curve (but see previous comment), and this causes a temporary negative value. What would be suddenly different about vegetation at this point; major vegetation changes over the catchment would take 10s to 100s of years to develop? If anything, naturally reduced vegetation high elevations and extended glacial extents should have worked in the same direction. More likely there was a contemporaneous reduction in agricultural practice (not “in any case”, but one half of the explanation of the odd value).

This brings one to thinking about the noise inherent in any proxy data set from lake sediments. A brief discussion of potential sources of noise (e.g., a decadal variation in river floods affecting a given sampling point) would be helpful. Seeing references to flood deposits in the methods makes this even more important to be clear about.

We agree with this comment, which will enable us to clarify this part of the discussion. It is unlikely that vegetation played a significant role in the high-frequency evolution of the excess erosion signal recorded in the catchment because, as the reviewer points out, it would take tens or hundreds of years for major changes to occur in the catchment's vegetation. In agreement with this comment, we have therefore deleted this part of the sentence.

Thanks to the three samples that we have added, we have now two periods presenting values of less than 1 for the erosion excess. These two periods correspond to the two periods of major glacier advances over the last few thousand years and the two coldest and wettest periods in Europe over the last two thousand years. The combination of a cold, wet climate and maximum glacial advance (Nussbaumer et al., 2012) may have encouraged erosion in the glaciated region. The great extent of glaciers may enhance the connectivity of glaciers with rivers and thus favour the sediment transport towards the lake as well.

These two periods are also known to have seen a marked decline in human activity, particularly in the mountains, due in part to climatic fluctuations at the time. The first period, known as the Migration Period, corresponds to one of the wettest and coldest periods in Western Europe for the past two millennia (Büntgen et al., 2011). This period was also marked by a geopolitical context that did not favour the development of human activities, particularly in the Alps, with invasions by tribes from the east following the collapse of the Roman Empire (Büntgen et al., 2011). Both the rapid climatic changes and the frequent epidemics had the overall capacity to disrupt the food production of agrarian societies at that time (Büntgen et al., 2011). Regarding the second period, the Little Ice Age (LIA), although several studies suggest an increase in human settlements and economic activities during this period in the western European Alps, the intensity of activity is reduced, especially when compared with previous peaks. Different practices or a summering period reduced by the colder conditions of the LIA (Büntgen et al. 2006) could explain this apparent contradiction (Giguët-Covex et al., 2023).

Accordingly, we modified the text to improve the discussion of these two periods (lines 189 – 196): *“These two periods were characterised by a transition to a colder and wetter climate than the average climate of the last two millennia. This had two effects: (i) increasing the surface area covered by glaciers, (ii) and enhancing erosion in the glacial region through the combined effect of increased precipitation and the advance of the glaciers. These two periods are also known to have seen a marked decline in human activity, linked to invasions during the Migration Period and to the reduction in agricultural practices probably induced, for both periods by the shortening of summer. These combined effects explain the strong decrease observed in the erosion excess signal for these two periods (Fig.2c).”*

Like any proxy, and like any archive, the use of lake sediments (and the method we have applied to them) presents biases and noise inherent in its very nature. To make our interpretation as robust as possible, we have carried out extensive error propagation work on our final record, combining each potential error associated with each level of interpretation and each assumption. It is true, however, that no error is presented on the excess erosion signal. To help the reader understand which variations are significant, we have decided to specify that variations in the excess erosion signal that do not exceed the variability observed over the correlation period (9.440 - 3.840 yr cal BP) will not be discussed, as they may be linked to the noise inherent in the archive used (lines 172 – 174): *“Variations in the signal of erosion excess that do not exceed the variability observed over the correlation period (9.440 - 3.840 yr cal BP) will not be discussed, as such variations may be linked to the noise inherent in the archive used.”*

151 et seq. When there is more erosion, more material, including various disturbance proxies, enter the lake (flux rate goes up). If the erosion were caused by deteriorating climate conditions, not increased human activity, it could appear that there had been more human activity when in fact there had not—if erosion caused a greater proportion of available marker proxies to be transported, thus maintaining or increasing concentrations, it will appear that flux goes up. This circularity is not really addressed in ref 24 either. One needs a fairly robust independent climate-change record to circumvent this possible circularity in the argument. The narrative climate-change background used here is, I suspect, the best there is. The LIA peak event may be the best illustration that climate deterioration and massive erosion in the mountains plus climate deterioration in the lowlands is not linked to enhanced lowland fluxes, though the human-disturbance figure shows this less clearly as the time-slices are variable. At any rate, as with other rather large assumptions, this should be addressed in the main text somewhere.

It is true that at the first order an increase in erosion will enhance the transport of environmental DNA into the lake and therefore exaggerate the agro-pastoral activity index derived from this particular proxy. However, in the upper catchment

study (mid & high altitude ranges), local erosion rates were also calculated and there are periods where the human activity index increases but erosion does not (see Giguët-Covex et al., 2023, supplementary information).

There are also sites where erosion does not fluctuate much, but where fluctuations in human activity are still visible. Unfortunately, this problem is inherent in the archives chosen and is difficult to overcome. However, most of the variations in activity have all been observed on a larger scale (on several sites, with different erosion dynamics due to the different geological and topographical contexts and the sensitivity of the vegetation cover). Integrating several sites with different dynamics is a way of circumventing these problems. There are also validations based on archaeological and historical data, which are completely independent (see Giguët-Covex et al., 2023 and references therein).

In addition, on many records, particularly at low and medium altitude sites, we are fortunate to have both DNA and pollen data. In these cases, the evolution of the intensity of human activity is similar if one considers pollen or DNA separately, and given that pollen and DNA do not have the same transfer processes: as pollen is mainly transported by the air, it is not influenced by the increase in erosion, unlike DNA; this is a compelling argument for stating that there is no significant effect of erosion fluctuations on the index obtained. The results of each analytical steps will be made available online.

According to this comment we decided to add some more information in the main text of the manuscript.

First, we decided to specify that the selected sites are considered as good illustrators of the human activities occurring in their altitudinal ranges (lines 206 – 212): *“We hypothesize that the two to three sites selected by altitudinal range illustrate the intensity of pastoral and/or agricultural activities in these areas. This assumption is considered as robust as the history of activities recorded at the sites in a given altitudinal range are very similar (Supplementary Fig. 8). Because the Lake Bourget integrates the sum of erosion products over all these altitudinal ranges (and is therefore not impacted by local physical and geological characteristics leading to variability in local erosion response), we can assess and quantify the extent of human impact on erosion at such large scale.”*

Then, we have also added a thorough discussion of the potential biases linked to each proxy used to reconstruct the index of agro-pastoral activities (lines 218 – 226): *“The integration of data from different lakes and different data-types allows us to offset specific taphonomic issues associated with each proxy and lake. Sedimentary DNA and coprophilous fungi signals are positively affected by climatic or human-triggered erosion, potentially leading to an overestimation of human activities, while pollen counting can be affected by a non-local origin of pollen grains and thus potentially reflect activities in other altitudinal ranges. However, for DNA and coprophilous fungi, because each lake catchment has its own sensitivity to erosion processes, potential biases are not expected to have had an identical pattern from one site to another. The binning of the data by chrono-cultural periods is an additional means of reducing noise in the signals generated by potential taphonomic problems.”*

Minor comments

11 - Better to say ...“loss of soil caused by a drastic increase in erosion”...otherwise one wonders exactly what is meant by “land”, a word that could be interpreted in many ways

The mention of “land” has been removed according to both this comment and the comment of Reviewer #1. The sentence has been modified into: *“A major feature of the Anthropocene is the drastic increase in global soil erosion.”*

13 - use of “the Earth system” is probably more common and clear

This change has been applied.

16 - to estimate quantitatively

This change has also been applied.

22 – Bronze Age onward and ... (no comma)

We modified the text accordingly.

25 – unclear what “stakes” means; “current major environmental issues”...?

According to this comment and the comment of Reviewer #1 we modified this into “environmental issues”.

27 - consistency: Human or human (latter preferable)

Thank you for this remark, we modified into human and checked throughout the entire manuscript the consistency.

28 -- increasing extent?

This change has been applied.

29 – not sure what “artificialized” means, please clarify

To clarify, we modified into “urbanized”.

35 -- experiments?

We modified “experiment” into “experiments”.

37 -- thereby evidenced

We replaced “hence” by “thereby”.

39 -- quantify accurately

This change has been applied.

40 -- ? beyond decades or, at most, a few centuries?

We modified the end of the sentence into : “beyond decades or, at most, a few centuries”.

48 – omit comma

The comma line 48 has been removed.

56 – great figure, but in caption, it would be helpful to say which is which – “The signal from the glaciated region corresponds....while that from the unglaciated....”

According to this comment we modified the sentence into: “*The signal from the glaciated region (in blue) corresponds to climate-driven erosion only, while the second one (in orange) is influenced by both climate and human activities (see Methods for detail).*”.

59 -- 10,000-year

Ok, we added a dash between “10,000” and “years-long”.

65 -- well suited (no hyphen, WORD is wrong!)

This change has been applied.

69 -- situated between?

We replaced “comprised” by “situated”.

76 -- omit comma

This change has been applied.

77 – they would affect

We replaced “it” by “they”.

81 – fix “sediment sediments”

We removed “sediment”.

91 – Phrase “Thanks to our approach” can be omitted. Ok to say “we obtained”..

We removed “Thanks to our approach (Fig. 1; Method), we obtained...” and replaced that by “Here, we obtained...”.

95 - associated with

This change has been applied.

103 Figure 2. Colour shading on left doesn’t have a scale and just shows in a quantitative way how far a point is from 1:1 ; it might be clearer with just joined dots and no shading, as the difference between the blue zone and the orange zone is obvious. Graphs should be labelled ABC here, as it is important to direct reader to the correct graph in the related discussion.

Thank you for this remark. We removed the shading for the blue zone on the bottom left panel. ABC labels were added and reference to them were made within the main text.

106 Colour shading on right needs clarification in caption....and the effect of human activities on erosion in the non-glaciated region, computed as a ratio of the measured erosion rate and that expected in the absence of human impact (right-hand scale, bottom graph/Fig 2C). Also “ER” for end of Roman period is confusing, as “E” is elsewhere used to signify “early”; use Post or After?

According to this comment, we added some clarification in the caption of Figure 2: “Colour shading of erosion signals in glaciated and non-glaciated regions highlights increases in intensity compared with early Holocene values. On panel c, the magnification scale represents the effect of human activities on erosion compared with the theoretical value calculated for erosion impacted solely by climatic fluctuations.”. We also replaced “End of Roman Period” by “Post Roman Period” and modify accordingly Fig 3.

111 – late Middle age (not latte)

This change has been applied.

136 – advance of glaciers led to.....

This change has also been applied.

171 – this statement is rather unclear.

Either human activities were adapted to lessen soil erosion, or the limits of erodible soil were reached, as previously proposed?

For greater clarity this sentence was modified into: *“This can correspond to a “system adjustment” phase during which human activities have deliberately evolved to lessen soil erosion or the limits of erodible soil were reached, as previously proposed...”*.

182 – finesse? Might need to be explained more

Perhaps...” This study highlights the importance of the retrospective approach for understanding the effect of human activities. Only this approach, based on multi-proxy analyses and long timescales, can deliver detailed information on the compounded effects of climate and human activities on soil erosion.

Thank you for this comment, we modified the sentence accordingly: *“This study highlights the importance of the retrospective approach for understanding the effect of human activities. Only this type of approach, based on multi-proxy analyses and long timescales, can deliver detailed information on the compounded effects of climate and human activities on soil erosion”*.

304 – 1-m offset

This change has been applied.

306 – in the overlapping

This change has also been applied.

314 – flood deposits

This correction has been made.

317 – see main text, reader should know about the age-model uncertainty

According to this comment we added in the main text (see lines 119-121) information about the age-model uncertainty: *“Thanks to a precise age model (Supplementary Fig. 4), with a median uncertainty of 60 years and a maximum uncertainty of no more than 100 years over the last 6,700 years, it is possible to accurately describe the evolution of these erosion signals over time.”*

325 – I think I can see why flood deposits are used, but it would be good to explain it here.

Some explanations have been added here: *“Flood deposits were selected as it makes it possible to obtain fine sediment samples covering a long period and representative of each sub-basin”*.

391 – replace 1/ and 2/ with 1) and 2)

Thank you for this remark, to keep consistency with the main text, we modified 1/ and 2/ into i) and ii).

398 – “to that sim” ??makes no sense. “To that end”?

This change has been applied.

404 – shifts in the relation of the correlation in erosion between.....

This change has been applied.

Reviewer #3 (Remarks to the Author):

This paper constitutes an important source of insights into the evolving landscape of soil erosion within the Alps, which is a prominent hotspot for soil loss in Europe. The authors' extensive work on an approximately 11-meter composite record to estimate erosion rates spanning several millennia is commendable. The meticulous detailing of their methods, including a new interesting source-to-sink approach, adds credibility to their findings. Moreover, the robustness of the age model, spanning roughly 9.4 thousand years before present, is particularly impressive.

I believe that the results presented in this manuscript hold significant value for publication in Nature Communications, particularly due to the previously unrecorded surge in soil erosion that commenced approximately 3800 years ago.

My main concern is the following:

The observed increase in erosion rates in both glaciated and non-glaciated regions around 3800 years ago (or 4.2k; see last sentence). In the glaciated region, sedimentation yield clearly demonstrates a step change during this period, as also indicated by ϵNd values. On the other hand, the non-glaciated region exhibits more of a

gradual upward trend, but both regions unmistakably indicate an increase around 3.8kBP. Given that human activity did not significantly impact the glaciated region, it seems that more discussions for this step change in sediment yield at 3.8kBP in the glaciated area be warranted. It would be interesting to explore into how the neoglacial cooling, a period marked by declining temperatures, could have affected sedimentation rates in both glaciated and non-glaciated regions. Perhaps a simple insolation curve in Fig.2 would give a better insight on this. Also, looking at Fig. 2's chronology, it makes me wonder if the step change actually occurred around 4.2k, aka the so-called "4.2k event"? As the age model is quite solid at this depth. Please provide more discussions.

Thank you for your comment. With regard to the difference in the upward trend that you mention between the erosion signal for the glaciated region and that for the non-glaciated region, we argue that we must not lose sight of the fact that the measurement errors on the signals presented here, particularly the one concerning erosion in the glaciated region, are significant, especially on points dated after 4,331 cal BP (see Fig. 2b). Once the error bars have been considered, it is not possible to describe and discuss the difference in trend observed over this specific period.

Thanks to this comment we improved the discussion of this first erosion surge (lines 124 – 128): *“During this period, called the Neoglacial period, the Alps experienced a transition towards a wetter and colder period compared to the Early Holocene. This change in climate led to glacier advance at high altitudes. The first phase of increase in the erosion signal being similarly observed in both regions is therefore linked to the effect of one of the main climatic transitions of the Holocene in the European Alps.”* Moreover, it is true that we did not make any connection between the Neoglacial period and the 4.2 ka BP event in our manuscript. However, even if it is likely that the two phenomena are linked here, we do observe only a long-term transition between a dry, warm period when erosion rates are low and constant and a colder, wetter period which leads to an initial increase in erosion and not a transient event with a return to a normal state. This transition occurred exactly between 4,893 and 4,066 yr cal BP. With only one value available between the two dates mentioned above (i.e., at 4,331 yr cal BP) it is not possible to relate this transition to the 4.2 ka BP event. This is why we have not included a reference to event 4.2 in our main text. Furthermore, the climatic phenomena at the origin of the Neoglacial period are not at the core of the discussion of the work we are presenting here, and so as not to complicate our message we have elected not to add an insolation curve to Figure 2.

Furthermore, we fully agree with the reviewer that it would be very interesting to discuss the different responses of glaciated and unglaciated regions during the Neoglacial. However, our record does not have sufficient resolution for such a discussion, but we keep this idea in mind for future work, for example by collecting new samples covering this period, to gain a better understanding of the evolution of erosion over this specific period.

Minor comments:

- Is any population size estimates for the different cultural epochs? Please add reference for each of these periods.

Thank you for your comment. Indeed, linking our indexes of agro-pastoral activities to estimates of population size would have been very useful and could have enabled us to discuss the effect of population growth on the erosion signal. However, no estimates of population size in the catchment or in the western European Alps are available. To our knowledge, a study on this specific subject is currently underway and should provide information on the evolution of population size in the western European Alps. However, this work is not yet complete.

- No grain-size analysis is shown in the paper. Has it been done?

Indeed, no grain-size distribution data is presented in this paper. The only point for which this information could have been important is if the fluctuations in grain size had an effect on the isotopic signature of neodymium, which in this case could have impacted our signal and therefore our mixing model. However, it has been shown on several occasions (Bayon et al., 2015 and references therein) that this was not the case. Some grain size analyses have been carried out to characterise deposits related to instantaneous events present in the sedimentary sequence and are therefore not presented in this manuscript. Several studies carried out on Lake Bourget have already presented grain-size data for other sedimentary sequences and have shown that there has been little change in grain-size over the last 10,000 years (e.g., Arnaud, 2005; Arnaud et al., 2012; Giguet-Covex et al., 2010). Typically, the grain size distribution shows a mode reflecting the detrital input and another one a biogenic calcite component. The only changes occurred during the recent eutrophication period (post-1930), but this period was not examined as part of this study.

References cited in this comment:

- Arnaud, F., 2005. Discriminating bio-induced and detrital sedimentary processes from particle size distribution of carbonates and non-carbonates in hard water lake sediments. *J. Paleolimnol.* 34, 519–526.
- Arnaud, F., Révillon, S., Debret, M., Revel, M., Chapron, E., Jacob, J., Giguet-Covex, C., Poulenard, J., Magny, M., 2012. Lake Bourget regional erosion patterns reconstruction reveals Holocene NW European Alps soil evolution and paleohydrology. *Quat. Sci. Rev.* 51, 81–92. <https://doi.org/10.1016/j.quascirev.2012.07.025>

Giguet-Covex, C., Arnaud, F., Poulenard, J., Enters, D., Reyss, J.-L., Millet, L., Lazzaroto, J., Vidal, O., 2010. Sedimentological and geochemical records of past trophic state and hypolimnetic anoxia in large, hard-water Lake Bourget, French Alps. *J. Paleolimnol.* 43, 171–190. <https://doi.org/10.1007/s10933-009-9324-9>

Line 81 : sediment sediments

According to this comment and the one of Reviewer #2 we removed “sediment”.

Line 149 : please provide the number of sites

We added information about the number of sites in this sentence.

Line 154 : provide more info about the standardization procedure

We changed Lines 215-218 for: *“The standardization consists in the normalization to 0-100 % of the values of all proxies. Once obtained for each lake, these normalized values are integrated in the calculation of an average index by lake. Finally, these indexes are group together by altitudinal range and for each chrono-cultural period (Fig 3).”*

Line 398 : aim, remove "s"

Here, “aim” was replaced by “end” according to this comment and the remark of Reviewer #2.

Line 412 : please provide more info (0-100%?)

According to this comment, we added some more information: *“In details, this standardization procedure consists of changing the highest value of each proxy to 100 % and calculating the other values proportionally.”*

REVIEWERS' COMMENTS

Reviewer #1 (Remarks to the Author):

I enjoyed reading this revised version. I consider the authors to have responded thoughtfully to my concerns. I'm still dubious about using so many named chrono-cultural periods in the text and especially on Figure 3, but the authors put forward a fair case for why they have chosen to stick with their use. One idea for Figure 3 would be to rotate the text so that the names of the periods can be spelled out in full? Or make the text boxes taller so the names of each period can be spelled out in full across multiple lines?

Reviewer #2 (Remarks to the Author):

I appreciate the authors' thoughtful commentary on the reviews and their approach to revision of this manuscript, which has improved in the key areas that the reviewers mentioned. The chronological control, the interplay of different processes at different elevations, and the conceptual explanation of why the lake-sediment record can be interpreted in the way done here are now clarified. I understand the argument about the naming of periods in Figs 2 and 3 and accept that is a reasonable way to do it.

Reviewer #3 (Remarks to the Author):

The manuscript's clarity has notably benefited from the revisions, and I commend the authors for their comprehensive responses to the raised points. Overall, I consider this study to be very well detailed, methodologically sound and it lays a solid foundation for future research endeavors dedicated to the challenging task of disentangling the effects of human activity from climate forcing. This contribution holds significant importance for the scientific community, especially paleolimnologists.

I only have minor comments as described below.

As also pointed out by the other reviewers, the assumptions made in the study are well-justified. The assumption of uniform behavior in both glacierized and non-glacierized regions in response to climate variability is probably correct. A suggestion would be to simply add temperature and precipitation data (in the supplementary materials) during the instrumental period for both glacierized and non-glacierized regions (if such data is available for the glacierized region, of course), and show their correlation. This may support the case for uniform responses through time.

Fig. 1: it would be helpful to have a map of the site showing Europe like for Supp. Fig. 2.

Fig. 2: it would be easier for the more general readers (as also points out by ref #1) to add a X-axis showing years in Common Era (CE), especially since you cite periods with great influence on the population (i.e., Black Death).

Please revise the order of the supp figures in the main text. Line 63: Supplementary 4 should not be before Supplementary 1.

Please verify these typos to stay consistent throughout the text: characterized vs characterised. Same for other word; synthesized vs synthesised.

Lines 246-248: please rewrite.

Nature Communications manuscript NCOMMS-23-31162-A

REVIEWER COMMENTS

Reviewer #1 (Remarks to the Author):

I enjoyed reading this revised version. I consider the authors to have responded thoughtfully to my concerns. I'm still dubious about using so many named chrono-cultural periods in the text and especially on Figure 3, but the authors put forward a fair case for why they have chosen to stick with their use. One idea for Figure 3 would be to rotate the text so that the names of the periods can be spelled out in full? Or make the text boxes taller so the names of each period can be spelled out in full across multiple lines?

We would like to thank the reviewer for the time spent evaluating our work and for the positive comments that helped improve this manuscript. We agree with this last comment and have therefore modified the boxes for the chrono-cultural periods in Figures 2 and 3 by enlarging them, which now allows the names of the periods to be spelled out in full for almost all the periods. We believe that the new versions of Figures 2 and 3 are now more readable.

Reviewer #2 (Remarks to the Author):

I appreciate the authors' thoughtful commentary on the reviews and their approach to revision of this manuscript, which has improved in the key areas that the reviewers mentioned. The chronological control, the interplay of different processes at different elevations, and the conceptual explanation of why the lake-sediment record can be interpreted in the way done here are now clarified. I understand the argument about the naming of periods in Figs 2 and 3 and accept that is a reasonable way to do it.

We would like to thank the reviewer for the time spent evaluating our work and for the positive comments that helped improve this manuscript throughout the review process.

Reviewer #3 (Remarks to the Author):

The manuscript's clarity has notably benefited from the revisions, and I commend the authors for their comprehensive responses to the raised points. Overall, I consider this study to be very well detailed, methodologically sound and it lays a solid foundation for future research endeavors dedicated to the challenging task of disentangling the effects of human activity from climate forcing. This contribution holds significant importance for the scientific community, especially paleolimnologists.

We would like to thank the reviewer for the time spent evaluating our work and for the positive comments that helped improve this manuscript throughout the review process.

I only have minor comments as described below.

As also pointed out by the other reviewers, the assumptions made in the study are well-justified. The assumption of uniform behavior in both glacierized and non-glacierized regions in response to climate variability is probably correct. A suggestion would be to simply add temperature and precipitation data (in the supplementary materials) during the instrumental period for both glacierized and non-glacierized regions (if such data is available for the glacierized region, of course), and show their correlation. This may support the case for uniform responses through time.

Unfortunately, the instrumental data for the study area only covers the period 1900-2023. We do not have any data for this period, so we cannot compare the two datasets. However, this is a very interesting comment, and we will try in the near future to work on the recent period in order to obtain erosion values that can be compared with the instrumental data.

Fig. 1: it would be helpful to have a map of the site showing Europe like for Supp. Fig. 2.

Thank you for this helpful comment. We modified the figure according to this comment by adding a map showing the location of the site in Europe.

Fig. 2: it would be easier for the more general readers (as also points out by ref #1) to add a X-axis showing years in Common Era (CE), especially since you cite periods with great influence on the population (i.e., Black Death).

We tried to add an X-axis showing years in Common Era on figures 2 and 3, but this did not simplify the readability of these figures. We believe, however, that the reader will see little inconvenience in the absence of such an axis, since whenever a calendar age is expressed in the text, its equivalent in year cal BP is also specified.

Please revise the order of the supp figures in the main text. Line 63: Supplementary 4 should not be before Supplementary 1.

As the text currently stands, it is difficult to list all the supplementary in order. However, particular care has been taken to ensure that all supplementary are mentioned at least once in the text.

Please verify these typos to stay consistent throughout the text: characterized vs characterised. Same for other word; synthesized vs synthesised.

Thank you for this comment. We have amended the text to reflect this comment.

Lines 246-248: please rewrite.

The typo was corrected.